# Exploiting LLMs for Automatic Hypothesis Assessment via a Logit-Based Calibrated Prior

**Yue Gong**[*]
Amazon Web Services
yuegongy@amazon.com

**Raul Castro Fernandez**
The University of Chicago
raulcf@uchicago.edu

## Abstract

As hypothesis generation becomes increasingly automated, a new bottleneck has emerged: hypothesis assessment. Modern systems can surface thousands of statistical relationships–correlations, trends, causal links–but offer little guidance on which ones are novel, non-trivial, or worthy of expert attention. In this work, we study the complementary problem to hypothesis generation: *automatic hypothesis assessment*. Specifically, we ask–given a large set of statistical relationships, can we automatically assess which ones are novel and worth further exploration? We focus on correlations as they are a common entry point in exploratory data analysis that often serve as the basis for forming deeper scientific or causal hypotheses.

To support automatic assessment, we propose to leverage the vast knowledge encoded in LLMs' weights to derive a prior distribution over the correlation value of a variable pair. If an LLM's prior expects the correlation value observed, then such correlation is not surprising, and vice versa. We propose the *Logit-based Calibrated Prior*, an LLM-elicited correlation prior that transforms the model's raw output logits into a calibrated, continuous predictive distribution over correlation values. We evaluate the prior on a benchmark of 2,096 real-world variable pairs and it achieves a sign accuracy of 78.8%, a mean absolute error of 0.26, and 95% credible interval coverage of 89.2% in predicting Pearson correlation coefficient. It also outperforms a fine-tuned RoBERTa classifier in binary correlation prediction and achieves higher precision@K in hypothesis ranking. We further show that the prior generalizes to correlations not seen during LLM pretraining, reflecting context-sensitive reasoning rather than memorization.

## 1 Introduction

Generating hypotheses from large data repositories is quickly becoming easier. Modern data discovery systems [2, 8, 25, 21] can enumerate every statistical relationship across datasets, and LLMs can draft thousands of plausible ideas by mining literature and data [33, 31, 29]. What used to take a researcher weeks now happens in minutes. This ease of generation, however, introduces a new bottleneck: assessment. Experts are flooded with machine-suggested relationships–correlations, causal links, trends, anomalies–without a clear signal for which ones merit deeper investigation. Many of these relationships are trivial, redundant, or already well known, forcing human experts to sift through a long list just to find a few that are novel.

For example, as illustrated in Figure 1, a correlation discovery system has surfaced tens of thousands of correlated variable pairs, leaving human experts to manually filter out trivial or expected patterns using their prior knowledge. A strong correlation between `daily temperature` and `ice cream sales`, for instance, is intuitive and quickly dismissed. In contrast, a negative correlation between `household`

---

[*]Work done at the University of Chicago

`income` and `housing prices` might appear counterintuitive and warrant further scrutiny. This manual triage must be repeated across thousands of pairs to uncover truly novel or surprising correlations, making the process highly labor-intensive. One might hope that ranking correlations by magnitude could alleviate this burden. However, as shown in Figure 1 (left panel where variable pairs are ranked by $|r_{\text{obs}}|$), stronger correlations are not necessarily more surprising—in fact, they often reflect well-known or redundant relationships, a trend we further verify in Section 5.

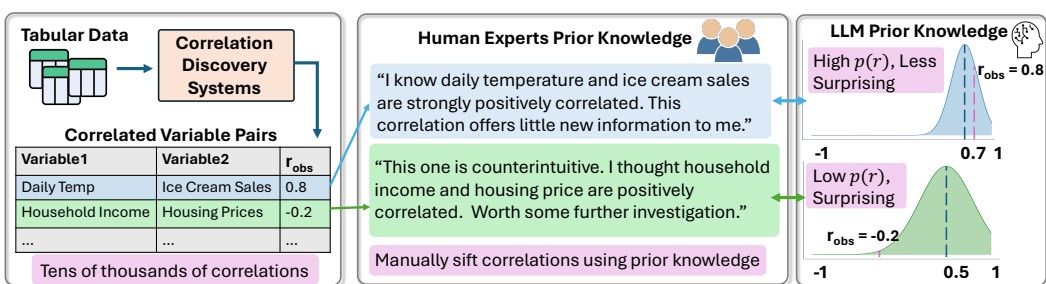

Figure 1: How Human Experts Assess Correlations Manually and How an LLM Can Help

In this paper, we study the complementary problem to hypothesis generation: *automated hypothesis assessment*. Specifically, we ask–given a large set of statistical relationships, can we automatically assess which ones are novel and worth further exploration? We focus on correlation relationships as a starting point, since they are a common entry point in exploratory data analysis and often serve as seeds for forming deeper scientific or causal hypotheses [11, 4].

To tackle this problem, we draw inspiration from how experts reason: they use prior knowledge to form expectations about a correlation's direction and magnitude. If the observed correlation ($r_{obs}$) matches expectations, it is unsurprising; if it deviates, it may signal something worth exploring. In essence, experts apply an implicit *prior* shaped by their knowledge and the variable context.

Our core idea is to approximate this human prior using the rich, encoded knowledge within LLMs [27, 18]. Specifically, we define the *LLM-elicited correlation prior*, $p_{\text{LM}}(r_{X,Y} \mid \mathcal{C}_{X,Y})$, as a predictive distribution over correlation values $r_{X,Y}$ between a variable pair $X, Y$ conditioned on their context $\mathcal{C}_{X,Y}$, such as the description for each variable. By prompting the LLM with this context, we elicit its belief about the correlation values, treating these beliefs as a proxy for human expectations.

The LLM-elicited correlation prior helps identify which correlations are novel and worth expert attention. For instance, in Figure 1, $p_{\text{LM}}(r_{\text{Daily Temp, Ice Cream Sales}} \mid \mathcal{C})$ centers around 0.7, making an observed value of 0.8 unsurprising. In contrast, a correlation of -0.2 against a prior centered at 0.5 signals high surprise. This surprise-based scoring offers a scalable way to surface potentially insightful correlations. In our later evaluation (Section 5), we show that the LLM prior highlights expert-validated hypotheses from noisy urban data [19].

In this work, we propose *Logit-based Calibrated Prior*, an LLM-elicited correlation prior which transforms the LLM's raw output logits into a calibrated, continuous predictive distribution over correlation values (Section 2). But how do we evaluate its quality?

First, we assess accuracy: if the prior's mode reliably predicts the sign and magnitude of observed correlations, it suggests alignment with empirical patterns. Second, we evaluate information content. A strong prior should assign high likelihood to observed correlations, reducing their information content relative to an uninformative baseline (e.g., a uniform prior). When applied at scale, this indicates the prior captures real-world patterns, easing the burden on analysts. Third, we measure calibration–whether the prior's uncertainty reflects reality–using 95% credible interval coverage. This is crucial for decision-making: overconfident priors exaggerate surprise and risk misdirecting expert attention. Finally, we ask a deeper question: is the prior reasoning from context, or merely recalling memorized correlations based on variable names? To probe this, we introduce a novel evaluation based on *contextual contradiction* to disentangle these possibilities.

To support these goals, we construct a benchmark of 2,096 variable pairs with observed correlations. We evaluate predictive quality and information reduction (Section 4), hypothesis discovery utility (Section 5), and whether the prior reflects contextual reasoning or memorization (Section 6).

Results are promising: our *Logit-based Calibrated Prior* achieves 78.8% sign accuracy and a mean absolute error of 0.26 on Pearson correlation coefficients in the range $[-1, 1]$, with strong calibration–95% intervals covering 89.2% of observed values. It also reduces the average information content from 0.69 (uniform prior) to 0.27. Our method outperforms baselines, including uninformative priors, Gaussian priors from LLM-verbalized parameters, and a fine-tuned RoBERTa classifier [30]. It also achieves higher precision@K when retrieving meaningful correlations in noisy urban data. For instance, it highlights a link between bike dock density and community wealth, a hypothesis studied in prior work [7], while down-ranking obvious patterns like library visitors and book circulation. Finally, we show that the prior generalizes beyond correlations seen during pretraining.

These results show that LLMs encode informative prior beliefs about statistical relationships, demonstrating their potential to serve as proxies for hypothesis assessment–a task that currently relies on human expertise and is highly labor-intensive. Our work highlights a promising direction for leveraging LLMs to help experts navigate large hypothesis spaces and make novel discoveries.

## 2 Logit-based Calibrated Prior (LCP): Constructing a Continuous Correlation Prior from LLM Logits

In this section, we present the *Logit-based Calibrated Prior (LCP)*, a method for constructing the correlation prior, $p_{\text{LM}}(r_{X,Y} \mid \mathcal{C}_{X,Y})$ –a predictive distribution over correlation values $r_{X,Y}$ between a variable pair $X, Y$ conditioned on their context $\mathcal{C}_{X,Y}$, such as the description for each variable.

One way to elicit a distribution from an LLM is to have it parameterize a fixed form–e.g., modeling its belief over a correlation as a Gaussian by providing a mean and standard deviation. However, this approach relies on the assumption that the model's internal belief distribution conforms to the chosen parametric form, which is not the case in most cases. To test this, we conducted a chi-square goodness-of-fit analysis [28] on the LLM's output distributions for 2,096 correlations. The normality assumption was rejected in 2,095 cases at the 5% significance level, indicating that the LLM's beliefs are poorly approximated by a Gaussian fit (see Appendix A for details). Moreover, this parametric approach introduces additional complexity: While the original goal is to estimate a single correlation value, it requires estimating additional parameters whose values are themselves subject to error.

**Our approach.** Rather than asking the LLM to estimate parameters of a fixed distributional form, we directly prompt it to predict the correlation coefficient (see prompt in Appendix H.1), and construct a full distribution over possible correlation values. While one could obtain this distribution by sampling from the model, it would be computationally expensive. To address this, we propose a more efficient strategy by constructing the prior directly from the LLM's logits. This approach does not assume the distribution's shape and allows the model to focus on estimating the correlation between the variables.

We begin by constructing a discrete probability distribution from the model's raw token logits (Algorithm 1). Without loss of generality, we assume $r$ denotes Pearson's correlation coefficient, constrained to the range $[-1, 1]$. At each decoding step $t$, the language model produces a real-valued logit vector $\ell_t$, where each entry $\ell_t^{(i)}$ corresponds to a token in the vocabulary. These logits are converted into log-probabilities via the softmax function. For a selected token $v_t$ at position $t$, its log-probability is given by $\log p_t^{(v_t)} = \ell_t^{(v_t)} - \log \sum_j \exp(\ell_t^{(j)})$.

We design a prompt that elicits a structured scalar response, such as *{"coefficient": "<value>"}*. To extract the correlation value, we first identify the start and end positions of `"<value>"` in the output sequence (line 4). Starting from this token position, we extract the top-$k$ tokens at each subsequent decoding step (line 5). A complete numeric response, such as `"-0.69"`, is composed of a valid sequence of tokens–e.g., a sign, integer part, decimal point, and numeric suffix. For instance, at the numeric suffix token, the model might assign different probabilities to completions like 69, 60, or 70. To prevent length biases, we ensure that positive and negative numbers use the same number of tokens in their representation (see Appendix B).

We enumerate all token sequences (line 5), concatenate them into strings (line 6), cast them to float values (line 8), and compute their joint log-probabilities by summing the log-probabilities of each token in the sequence (line 12). We discard any sequences that produce invalid float values or values outside the valid correlation range $[-1, 1]$ (line 10). When multiple token sequences map to the same numeric value (e.g., `"0.65"` and `".65"`), we aggregate their unnormalized probabilities (line 13-17).

Finally, we normalize across all valid correlation values using the softmax function to obtain a discrete probability distribution, $\{(r_j, p_j)\}_{j=1}^N$ (line 19) where $r_j$ is a decoded correlation value, and $p_j$ is its model-assigned probability.

**Impact of Single-Path Decoding.** Our approach decodes one token at a time conditioned on the most likely token from the previous step. In particular, we first select the most probable sign token (e.g., + or -) and condition all subsequent decoding on that choice. This introduces an approximation: we compute the joint probability of each numeric string under a single sign path, rather than marginalizing over both. While a full multi-beam search would more faithfully capture the true joint distribution by exploring multiple branches at each step, we find that this approximation works well in practice. First, the model achieves *high sign accuracy*: as shown in Section 4, it predicts the correct sign 78.8% of the time, so most sequences are decoded under the correct branch. Second, it exhibits *high sign confidence*: across 2096 correlation predictions, the median probability gap between the two signs is 99.8% (77.7% on average), indicating that the probability mass of the alternative branch is negligible. Finally, our method ensures *scalability*: a full beam search would increase decoding cost exponentially with sequence length, whereas the single-path strategy scales efficiently to tens of thousands of variable pairs while maintaining strong empirical performance.

**Smoothing to Obtain a Continuous Prior.** The discrete distribution is sparse and limited to discrete values determined by the tokenizer and top-$k$ decoding strategy. However, downstream tasks–such as computing surprise in Figure 1–require probability density at arbitrary values. To support this, we smooth the distribution using a weighted sum of Gaussian kernels centered at each decoded value. Since Pearson correlations lie in $[-1, 1]$, we truncate and renormalize the distribution to ensure it integrates to one. The final LCP density function $f(r)$ is defined as:

$$f(r) = \frac{1}{Z} \sum_{j=1}^N p_j \cdot \frac{1}{\sqrt{2\pi\sigma^2}} \exp\left(-\frac{(r - r_j)^2}{2\sigma^2}\right), \quad r \in [-1, 1]$$

where $\sigma$ is the standard deviation of each kernel and controls the degree of smoothing, and $Z$ is the normalization constant.

---

**Algorithm 1** ConstructDiscretePriorFromLogits

---

1: **Input:** Token logits $\{\ell_t\}_{t=1}^T$, structured output template $\mathcal{T}$ such as *{"coefficient": "<value>"}*
2: **Output:** Discrete prior $\{(r_j, p_j)\}_{j=1}^N$
3: Initialize empty map: logp_map $\leftarrow \emptyset$
4: $t_0, t_1 \leftarrow$ FindValueTokenSpan($\{\ell_t\}_{t=1}^T, \mathcal{T}$)                 ▷ Locate start and end positions of the value field
5: **for all** sequences $s = (v_{t_0}, \ldots, v_{t_1})$ from top-$k$ tokens at each position **do**
6:     str $\leftarrow$ concat($s$)
7:     **if** is_valid_float(str) **and** float(str) $\in [-1, 1]$ **then**
8:         $r \leftarrow$ float(str)
9:     **else**
10:         **continue**
11:     **end if**
12:     $\log p_r \leftarrow \sum_{t=t_0}^{t_0+L} \log p_t^{(v_t)}$
13:     **if** $r \in$ logp_map **then**
14:         logp_map[$r$] $\leftarrow \log\left(\exp(\text{logp\_map}[r]) + \exp(\log p_r)\right)$
15:     **else**
16:         logp_map[$r$] $\leftarrow \log p_r$
17:     **end if**
18: **end for**
19: $\{(r_j, p_j)\}_{j=1}^N \leftarrow$ softmax(logp_map)                 ▷ Normalize log-probs into a valid probability distribution
20: **return** $\{(r_j, p_j)\}_{j=1}^N$

---

Selecting an appropriate kernel standard deviation $\sigma$ is critical to ensure the prior reflects realistic uncertainty. If $\sigma$ is too small, the resulting distribution will be overconfident and overly spiky; if too large, it will be underconfident and overly diffuse. Standard bandwidth selection rules, such as Scott's [23] or Silverman's rule [26], are not applicable in our setting, as they assume i.i.d. samples from an underlying distribution. In our case, in contrast, the discrete values and their probabilities are derived from LLM output logits and reflect model-specific beliefs, not empirical frequencies.

To address this, we tune $\sigma$ using a held-out validation set by minimizing the average negative log-likelihood at the observed correlation values:

$$\sigma^* = \arg\min_{\sigma} \; \mathbb{E}_{r_{\text{obs}} \sim \mathcal{D}_{\text{val}}} \left[ -\log p_{\sigma}(r_{\text{obs}}) \right],$$

This objective penalizes priors that assign low probability density to ground-truth correlations, thereby encouraging distributions that place probability mass closer to the observed values. Optimizing $\sigma$ this way calibrates uncertainty to reflect empirical variability and improves downstream reliability. The validation set $\mathcal{D}_{\text{val}}$ consists of 300 randomly sampled correlations, disjoint from our evaluation dataset. The optimized value $\sigma^* = 0.4$ is used for LCP.

The kernel standard deviation $\sigma$ does not need to be re-tuned as long as four key elements remain unchanged: the LLM, the prompting strategy, the task (predicting Pearson correlation coefficients), and the kernel function used. This is because $\sigma$ corrects for the systematic bias in the model's uncertainty–that is, whether the model tends to be consistently overconfident or underconfident in its predictions. When the model architecture, prompt design, task, and the kernel function remain fixed, this bias remains stable across inputs, even if individual predictions vary. In this setting, a single globally tuned $\sigma$ is sufficient to calibrate the model's uncertainty across a broad range of variable pairs. We further demonstrate in our evaluation (Section 4, 5) that the selected $\sigma$ generalizes well on the evaluation dataset, demonstrating its robustness. However, if any of these components change–such as switching to a different model, altering the prompt, targeting a different correlation metric, or switching to a different kernel function–the structure of the output distribution may shift, and $\sigma$ should be re-tuned to ensure proper calibration.

In the evaluation, we compare the Logit-based Calibrated Prior against two baseline methods for constructing correlation priors, highlighting the advantages of avoiding parametric assumptions and applying proper calibration. The first is a Gaussian prior, which assumes the LLM can directly parameterize a normal distribution by predicting its mean and standard deviation. The second is an uncalibrated KDE prior, which is similar to our method, but selects the kernel standard deviation using Scott's rule based on the empirical standard deviation of the discrete probability distribution.

## 3   Benchmark Construction

We curate a benchmark of 2,096 real-world variable pairs to evaluate correlation priors. Each entry includes two variables, their descriptions, a dataset summary, and the observed Pearson correlation $r_{\text{obs}} \in [-1, 1]$, computed from raw data. The benchmark combines variable pairs from the Cause-Effect Pairs [15] and Kaggle [30] datasets. We have open-sourced our code and data at `https://github.com/TheDataStation/LLM-Prior-for-Correlation-Assessment`.

The Cause-Effect dataset contains 108 variable pairs with known causal relationships. We retain 96 pairs where the correlation is statistically significant ($p < 0.05$). The Kaggle dataset consists of correlations between variable pairs extracted from publicly available tables on Kaggle. The original dataset provides variable names but lacks variable descriptions. To enrich the context for variables, we use the Kaggle API to retrieve dataset summaries and employ GPT-4o (see Appendix H.5) to assess whether the variable names are self-descriptive[2]. We filter out non-informative names (e.g., single characters or generic identifiers like "Unnamed: 0") and retain only those pairs for which both variables are judged meaningful. This further cleaning allows us to isolate and study the model's ability to reason about relationships, rather than its ability to interpret metadata.

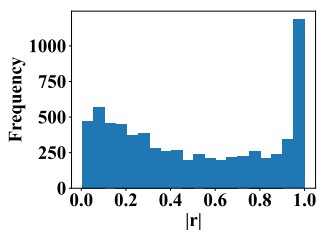

Figure 2: The Bias toward High Correlations in Kaggle dataset

After filtering, we obtain 7045 statistically significant correlations ($p < 0.05$). To mitigate the bias toward extreme correlations (see Fig. 2), we perform stratified sampling by $|r|$: divide the range $[-1, 1]$ into 10 equal-width bins and sample 200 correlations per bin, yielding a balanced set of 2,000.

---

[2]Kaggle API does not support retrieving variable descriptions

A balanced sample ensures fair evaluation across all correlation strengths, preventing the model's performance from being skewed by overrepresented low or high $|r|$ values.

## 4  How Well Does LCP Predict Empirical Correlations?

We evaluate LCP by measuring how well it predicts observed correlations. First, we assess *predictive accuracy* using two metrics: *sign accuracy*, the fraction where $\hat{r} \cdot r_{\text{obs}} > 0$, and *absolute error*, $|\hat{r} - r_{\text{obs}}|$, where $\hat{r}$ is the mode of the prior. We calculate the mode using grid sampling. Next, we evaluate differential information content by computing $-\log p(r_{\text{obs}})$, adapting Shannon's self-information [24] to the continuous case. For simplicity, we refer to it as *information content* hereafter. A good prior assigns high likelihood to observed values, reducing the information content of the corpus and easing analyst workload. Finally, we assess *calibration* by 95% credible interval coverage–the fraction of cases where $r_{\text{obs}}$ falls within the prior's 95% credible interval. Calibration is critical: an overconfident prior may exaggerate surprise from small deviations, leading to false positives and misleading experts.

We compare LCP with the following baselines. All methods use GPT-4o (2024-08-06) [17] as the underlying model.

• **Uniform Prior:** A non-informative baseline with constant density 0.5 over $[-1, 1]$. The sign accuracy for it is measured by randomly guessing the sign of the correlation.

• **Gaussian Prior:** We adapt the method from Capstick et al. [1], which elicits Gaussian priors via LLM-prompted mean and standard deviation, to model a truncated Gaussian prior over correlations in $[-1, 1]$ (see Appendix H.2).

• **KDE Prior:** A kernel density estimation using Gaussian kernels, where the kernel standard deviation $\sigma$ is set using a weighted version of Scott's rule: $\sigma = 1.06 \cdot \hat{\sigma} \cdot n_{\text{eff}}^{-1/5}$, where $\hat{\sigma}$ is the weighted standard deviation of $\{(r_j, p_j)\}$, and $n_{\text{eff}}$ is the effective sample size.

**Results.** Fig. 3 reports the average value of each metric across all correlations, positioned in a quadrant plot. Complementarily, Fig. 4 presents the full distributions of absolute error, $p(r_{\text{obs}})$, and information content. Fig. 3 shows that LCP achieves the best balance, matching the highest sign accuracy (78.8%) of KDE while providing significantly better calibration (89.2% coverage). In contrast, the uncalibrated KDE and Gaussian priors are overconfident, assigning low likelihood to $r_{\text{obs}}$ and yielding poor coverage (59.9% and 49.1%, respectively). On the other hand, the uniform prior offers high coverage (92.3%) but suffers from poor accuracy and high absolute error ($|\hat{r} - r_{\text{obs}}| = 0.51$).

In addition, LCP significantly reduces the average information content of the correlation corpus—from 0.69 under a uniform prior to 0.27, indicating that it assigns higher likelihood to observed correlations. In contrast, the Gaussian and KDE priors increase the average information content to 4.10 and 1.73, respectively, due to their overconfident predictions. This is reflected in the long tail of low-density values in Fig. 4b. As shown in Fig. 4, LCP yields more concentrated distributions for both likelihood $p(r_{\text{obs}})$ and information content, highlighting better calibration .

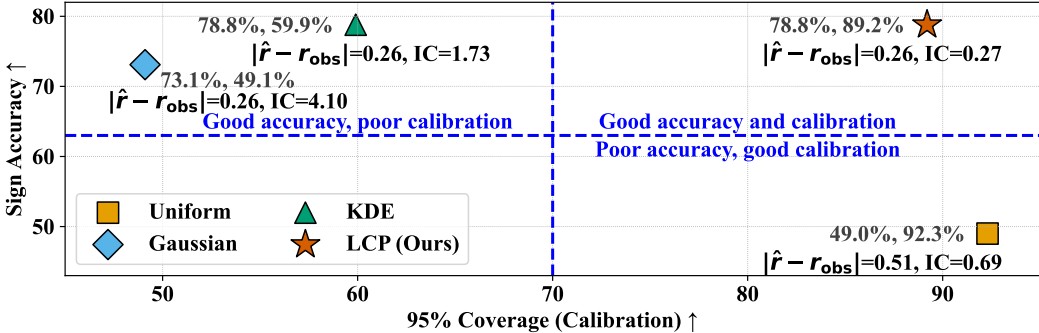

Figure 3: Accuracy vs. Calibration of Correlation Priors (IC=Information Content)

To understand the poor calibration of the Gaussian and KDE priors, we examine their kernel standard deviations. Both produce overly small $\sigma$ values, leading to sharply peaked densities. The median $\sigma$ is

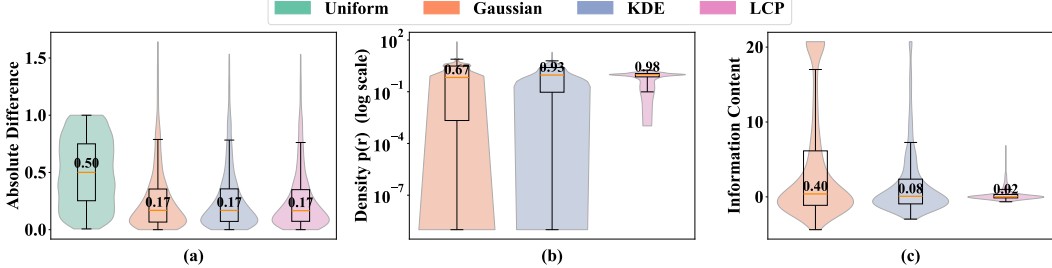

Figure 4: Full Distribution of Metrics over Different Priors

0.10 for the Gaussian prior and 0.08 for the KDE prior–both much smaller than the fixed $\sigma = 0.4$ in LCP. Under the Gaussian prior, the LLM returned $\sigma = 0.1$ in 74% (1,552/2,096) of cases. The full distribution of $\sigma$ values is shown in Appendix C. As we further examine in Appendix D, this behavior arises because the LLM interprets $\sigma$ as sampling variability and implicitly assumes a fixed sample size of 100—producing a default value of $\sigma = 0.1$ regardless of context. The predicted $\sigma$ captures expected variation from random sampling (*aleatoric uncertainty*), but fails to adjust based on the input context or account for uncertainty arising from limited knowledge (*epistemic uncertainty*) [12, 5].

Figure 5 analyzes LCP's behavior across ten bins of observed correlation $r_{\text{obs}}$. The bias $\hat{r} - r_{\text{obs}}$ decreases with $r_{\text{obs}}$: the model overestimates strong negatives and underestimates strong positives. Sign accuracy is lowest when $|r_{\text{obs}}|$ is small, bottoming out near $-0.15$ and rising sharply beyond $|r_{\text{obs}}| \gtrsim 0.3$, reaching near-perfect accuracy for $|r_{\text{obs}}| \geq 0.7$. In Fig. 5c,d, the prior assigns lowest density (i.e., highest information content) to moderately negative correlations ($r_{\text{obs}} \approx -0.5$), indicating weaker estimation. In contrast, strong positives receive the highest density and lowest information content. Overall, the prior shows asymmetric error: it performs best on strong positives and struggles with moderate negatives, consistently underestimating correlation magnitude—a reflection of the LLM's conservative predictions without direct data access.

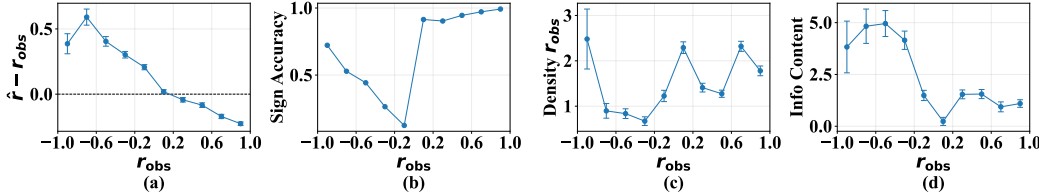

Figure 5: Performance across ten bins of the true correlation $r_{\text{obs}}$

**Comparison of LCP and RoBERTa on Binary Correlation Classification.** While LCP models a full distribution, BERT- and RoBERTa-based classifiers [14, 6] can be adapted for binary correlation prediction–determining whether a pair of variables is correlated based on a predefined threshold. We adopt the method from Trummer [30], who fine-tune RoBERTa using labeled pairs to build a correlation classifier. LCP is adapted to solve the binary classification task by thresholding its mode, enabling direct comparison with classification-based approaches.

To ensure a fair comparison, we first evaluate RoBERTa in a zero-shot setting, matching LCP, which requires no training. We then fine-tune RoBERTa on 20% of the benchmark, following standard practice for applying RoBERTa to downstream tasks. Figure 6 shows performance across different correlation thresholds. Note that RoBERTa must be re-trained for each threshold.

LCP consistently outperforms both baselines in terms of accuracy, F1, and MCC across all thresholds–achieving up to 0.84 accuracy, 0.79 F1, and 0.53 MCC–despite being entirely zero-shot. This indicates that our method provides the most balanced predictions overall. Zero-shot RoBERTa behaves like a one-class detector: it predicts correlated for every pair, yielding perfect recall but zero MCC and rapidly deteriorating accuracy/precision as the threshold tightens from 0.5 to 0.8. Fine-tuned RoBERTa corrects this imbalance to some extent after seeing 20% of the data, but its gains are threshold-specific and require retraining whenever the decision boundary moves. In contrast, By producing a full predictive distribution over $r$, LCP naturally adapts to different thresholds.

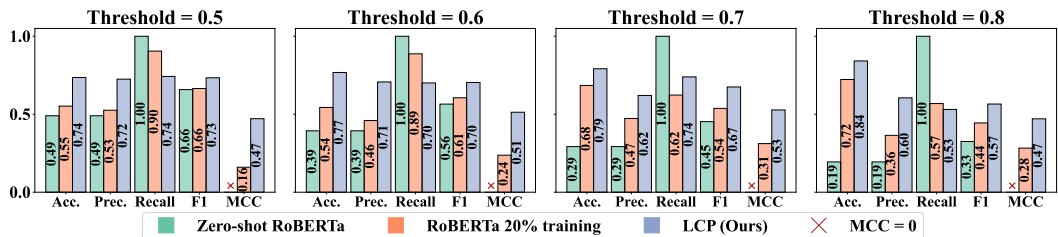

Figure 6: Classification Performance over Different Correlation thresholds

**Sensitivity Analysis of LCP.** We conducted a sensitivity analysis to assess the effect of two factors on LCP's robustness: (i) the choice of kernel function used for smoothing, and (ii) the validation set size used to calibrate the smoothing parameter $\sigma$.

We compared three alternative kernels-Uniform, Epanechnikov, and Triangle-against the Gaussian kernel used in our main results. For each kernel, we tuned the bandwidth parameter $\sigma$ on a held-out validation set and evaluated performance on 2096 correlated pairs. The results in Table 1 show that LCP maintains strong sign prediction accuracy and calibration across all kernel choices.

Table 1: Sensitivity to kernel function used for smoothing.

| Kernel | Sign Accuracy | $|\hat{r} - r_{obs}|$ | 95% Coverage |
|---|---|---|---|
| Uniform | 76.4% | 0.32 | 82.1% |
| Epanechnikov | 78.8% | 0.26 | 89.7% |
| Triangle | 78.8% | 0.31 | 90.9% |
| Gaussian (reference) | 78.8% | 0.26 | 89.2% |

These findings suggest that LCP's performance is not sensitive to the specific choice of kernel function, as long as the bandwidth is appropriately tuned. However, due to differences in kernel shape, the optimal $\sigma$ should be re-selected using a validation set when switching kernels.

Table 2: Optimal $\sigma$ across validation set sizes.

| Set Size | 300 | 500 | 1000 | 2000 |
|---|---|---|---|---|
| $\sigma_{optimal}$ | 0.40 | 0.38 | 0.38 | 0.41 |

We also evaluated how the optimal smoothing bandwidth $\sigma$ varies with the size of the held-out validation set. As shown in Table 2, the optimal $\sigma$ remains stable across different validation sizes, indicating that LCP is robust to sample size variations.

## 5 Using LCP to Retrieve Expert-Flagged, Hypothesis-Worthy Correlations

Can LCP support hypothesis assessment in noisy, real-world settings? We evaluate it on Nexus [8], a system designed to help domain experts discover correlations in urban data. Nexus computes 40,538 pairwise correlations from Chicago Open Data [19] by aligning and aggregating numeric attributes from different tables–either temporally (by month) or spatially (by census tract). Attributes are summarized (e.g., via mean or sum), joined on a shared key, and then correlated. This pipeline introduces real-world challenges: joins across sources, aggregation choices, and missing values–all of which impact the resulting correlations.

Of the full set, 15 correlations were labeled as hypothesis-worthy by human experts in the original Nexus evaluation. For example, a correlation between bike dock density and community wealth suggests stations are more common in affluent areas, a hypothesis studied in [7]. We use these expert-flagged examples to evaluate how well LCP retrieves hypothesis-worthy correlations in messy, transformed data. Since ground-truth correlation values are unavailable due to data aggregation and imputation, we adopt an information retrieval setup: treating the 15 expert-flagged correlations as targets within a pool of 115, formed by adding 100 random samples from the full Nexus corpus.

We compare five ranking strategies: (i) random, (ii) by absolute correlation $|r|$, (iii) by increasing probability assigned by a RoBERTa model fine-tuned on 20% of the benchmark, where lower

probability of the "correlated" class indicates higher surprise, (iv) by LLM surprise, where the LLM (GPT-4o) is given the full metadata (column names, table names, description, and observed correlation) and prompted to classify the correlation as "surprising" or "not surprising", and (v) by increasing prior likelihood $p(r_{obs})$ under LCP, treating lower likelihood as more surprising. We report Precision@5, @10, @15, and the average rank of expert-labeled correlations.

Table 3: Retrieval Performance Comparison

| Method | Precision@5 | Precision@10 | Precision@15 | Average Rank ↓ |
|---|---|---|---|---|
| Random Ranking | 0.13 | 0.13 | 0.13 | 58.0 |
| Ranked by $|r|$ | 0 | 0 | 0 | 95.4 |
| Ranked by RoBERTa | **0.60** | 0.60 | 0.53 | 30.9 |
| Ranked by LLM Surprise | 0.4 | 0.2 | 0.13 | 29.1 |
| Ranked by LCP | **0.60** | **0.80** | **0.60** | **21.5** |

As shown in Table 3, ranking correlations by LCP outperforms all baselines–achieving up to 0.80 Precision@10 and reducing the average rank of expert-labeled correlations to 21.5, compared to 30.9, 58.0 and 95.4 for the RoBERTa, random and $|r|$-based rankings, respectively (see Appendix E for a derivation of the expected performance under random ranking). Ranking by $|r|$ performs worst, with the highest average rank and zero precision, as extreme correlations often reflect trivial or redundant relationships (e.g., repeated attributes across years), not meaningful insights in Chicago Open Data.

Using LCP, all four correlations related to the expert-labeled hypothesis—that bike stations are more likely to be located in wealthier areas—are ranked within the top 6. In contrast, an unsurprising correlation–the one between library visitors and library circulation–is ranked much lower at 95th. These results demonstrate that LCP can surface correlations that align with expert judgment, even in the presence of data transformations and noise.

# 6   Is LCP Reasoning from Context or Relying on Memorization?

We evaluate whether LCP is *reasoning from context* or simply relying on *memorization*, a crucial distinction for generalization beyond the model's pretraining data. This is essential for hypothesis assessment, where many relationships are unseen during training and depend on context. To probe this, we introduce an evaluation based on *contextual contradiction*. For each variable pair, we construct an alternate context that plausibly reverses the original correlation, simulating a counterfactual. We then re-derive the prior by prompting the LLM with this modified context (Appendix H.4). If the model adjusts its belief accordingly, it suggests reasoning from context rather than memorization.

**Contradictory Context Generation.** We use the Cause-Effect Pairs dataset to construct counterfactual scenarios. From this dataset, we select 84 variable pairs where the model initially predicts the correct correlation sign. For each pair, we prompt Gemini 2.5 Pro to generate a new context that plausibly reverses the original relationship (see Appendix H.3). All 84 generated contexts are manually reviewed by the authors to ensure the reversal is logically sound and free of explicit cues (e.g., phrases like "therefore there should be a negative correlation"). We assign the negated correlation $-r_{obs}$ as the new observed value. Since these contexts are synthetic and no real data exists, these new $r_{obs}$ values serve as approximations.

**Result.** Table 4 shows the performance of correlation priors on reversed correlations. LCP achieves 100% sign accuracy on the original contexts, dropping slightly to 95.2% under contradictory contexts. Manual inspection reveals that two of the four errors stem from reasoning failures: the model grasps the high-level logic but fails at the final inference step in multi-hop scenarios (see Appendix F). LCP also maintains strong calibration, with 92.9% coverage at the 95% level, and achieves lower information content (0.25 vs. 0.69) and absolute error (0.30 vs. 0.55) compared to the uniform prior.

Table 4: Performance of correlation priors on correlations with contradictory contexts.

| Method | Sign Acc. (↑) | $|\hat{r} - r_{obs}|$ (↓) | Information Content (↓) | 95% Coverage (↑) |
|---|---|---|---|---|
| Uniform | 0.464 | $0.55 \pm 0.25$ | $0.69 \pm 0.00$ | 92.3% |
| LCP (ours) | **0.952** | **$0.30 \pm 0.28$** | **$0.25 \pm 0.98$** | **92.9%** |

This experiment shows that LCP is not merely recalling memorized correlations. When given counterfactual contexts, it updates its predictions accordingly, achieving 95.2% sign accuracy with strong calibration and low error. These results suggest that LCP generalizes beyond pretraining and behaves dynamically, a crucial property for real-world hypothesis assessment.

## 7  Related Work

**Elicit Priors from Human Experts.** O'Hagan et al. [16] and Gosling [9] introduce the SHELF framework, a structured protocol for eliciting expert judgments and converting them into probability distributions. The process involves training, individual assessments, group discussions, and consensus-building, followed by fitting a statistical distribution to the agreed-upon judgments. This human elicitation process is costly and time-consuming, whereas our approach exploits the rich knowledge encoded in LLM weights to approximate expert priors automatically.

**LLMs for Regression Tasks.** Several works exploit the knowledge encoded in LLMs for regression. Choi et al. [3] use LLMs for feature selection by prompting whether a variable is predictive of a given target, while others [20, 10, 1] aim to model prior distributions over feature weights. For example, Requeima et al. [20] require training examples to guide the LLM in generating output distributions, and Capstick et al. [1] assume the LLM can directly parameterize a distribution by prompting it to output means and standard deviations given feature and target names. In contrast, our work focuses on constructing a prior distribution over correlation coefficients between variable pairs before observing any data, using raw LLM logits directly—without requiring the model to parameterize a distribution.

**Additional Related Work.** We include further discussion on data discovery systems and automatic hypothesis generation in Appendix G.

## 8  Conclusions

In this paper, we propose the Logit-based Correlation Prior, an LLM-elicited prior that transforms raw output logits into a calibrated, continuous predictive distribution over correlation values—paving the way for automatic hypothesis assessment. Our experiments show (i) LCP achieves the best balance between accuracy and calibration for predicting empirical correlations, outperforming Uniform, Gaussian, and KDE priors; (ii) LCP outperforms a fine-tuned RoBERTa classifier on binary correlation classification; (iii) LCP effectively highlights hypothesis-worthy correlations flagged by human experts in noisy urban data; and (iv) LCP goes beyond memorizing correlation values from pretraining, performing contextual reasoning.

## 9  Limitations

Generating an LCP requires an LLM call per correlation, which can be costly at scale. To improve scalability, preprocessing steps—such as filtering out redundant variable pairs across similar datasets—can help reduce the number of required queries. LLMs may also produce false positives or negatives. An LLM may possess knowledge beyond that of human experts, causing it to dismiss correlations that are actually insightful to the experts (false negatives). It can also misinterpret well-known relationships, incorrectly flagging them as surprising (false positives), as shown in Appendix F.

## 10  Acknowledgments

We thank the anonymous reviewers for their constructive feedback, which significantly improved the clarity and quality of this work. This work was supported partially by the National Science Foundation (CAREER Award 2340034) and the Data Ecology Research Initiative at the Data Science Institute, University of Chicago.

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

# A    Normality Test via Chi-Square Goodness-of-Fit

To assess whether the LLM's output distribution over correlation values conforms to a Gaussian shape, we perform a chi-square goodness-of-fit test. For each correlation prompt, we obtain a discrete probability distribution $\{(r_j, p_j)\}_{j=1}^N$, where each $r_j \in [-1, 1]$ is a decoded numeric value and $p_j$ is the associated model-assigned probability mass, derived from token-level logits.

We convert the probability mass function into a set of pseudo-counts by assuming a nominal sample size $M = 1000$, yielding observed counts $O_j = M \cdot p_j$. We then estimate the mean $\mu$ and variance $\sigma^2$ of the distribution as follows:

$$\mu = \sum_{j=1}^N r_j \cdot p_j, \qquad \sigma^2 = \sum_{j=1}^N (r_j - \mu)^2 \cdot p_j.$$

Next, we compute the expected count for each support point $r_j$ under a fitted Gaussian:

$$q_j = \frac{1}{\sqrt{2\pi\sigma^2}} \exp\left(-\frac{(r_j - \mu)^2}{2\sigma^2}\right),$$

which we normalize to form a probability distribution $\tilde{p}_j = q_j / \sum_j q_j$, and then scale to expected counts $E_j = M \cdot \tilde{p}_j$.

The chi-square test statistic is computed as:

$$\chi^2 = \sum_{j=1}^N \frac{(O_j - E_j)^2}{E_j}.$$

The null hypothesis is that the observed distribution comes from the fitted Gaussian. We evaluate the $p$-value corresponding to the computed $\chi^2$ and reject the null at the 5% significance level.

Applied to the 2,096 correlations in our benchmark, the normality hypothesis was rejected in 2,095 cases, indicating that the LLM's output distributions are poorly approximated by a parametric Gaussian form. This result justifies our non-parametric approach, which avoids imposing a fixed distributional shape.

# B    Preventing Length Biases

In our setup that uses the GPT-4o tokenizer, both positive and negative correlation values are represented with the same number of tokens. Specifically, both forms include four tokens: sign, integer part, decimal point, and fractional part.

For example, the response:

```
{
    "coefficient": "0.5"
}
```

is tokenized as ' "', '0', '.', '5'.

While:

```
{
    "coefficient": "-0.6"
}
```

is tokenized as ' "-', '0', '.', '6'.

In our decoding process, we always treat the token immediately following the colon as the sign token, trimming the prefix (' "') before it. For positive numbers, the sign token can be an empty string or '+', while for negatives it is '-'. As a result, negative numbers do not suffer a disadvantage due to extra token length.

## C Distribution of Kernel Standard Deviations

Figure 7 shows the distribution of kernel standard deviations $\sigma$ used in the Gaussian and KDE priors. Both priors tend to produce small $\sigma$ values, contributing to overconfident and poorly calibrated predictions. The median $\sigma$ is 0.10 for the Gaussian prior and 0.08 for the KDE prior.

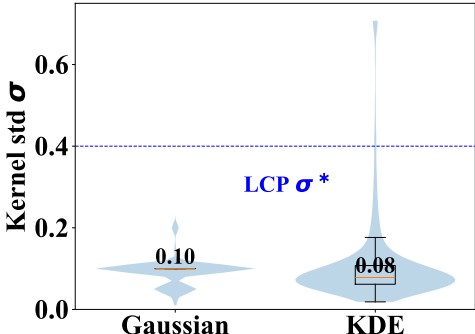

Figure 7: Distribution of kernel standard deviations for Gaussian and KDE priors.

## D Understanding the LLM's Behavior in Reporting Standard Deviations

The LLM (GPT-4o) favors the value 0.1 when reporting standard deviations. $\sigma = 0.1$ appears in 74% of cases (1,552 out of 2,096 prompts). To better understand this behavior, we conducted a targeted analysis of the LLM's internal assumptions when predicting $\sigma$.

We prompted GPT-4o with 50 column pairs whose names were random strings with no semantic meaning (e.g., abc123, xzy987). This design removes contextual cues, allowing us to observe the model's default behavior under maximum uncertainty. In all 50 cases, the predicted correlation coefficient was exactly zero, and in 46 out of 50 cases, the predicted standard deviation was 0.1. The strong preference for $\sigma = 0.1$ even in the absence of context suggests that, when prompted to express its uncertainty as the standard deviation of a normal distribution, the LLM may default to fixed assumptions—such as an implicit sample size—rather than adjusting its estimate based on contextual information.

To investigate why $\sigma = 0.1$ is so commonly predicted, we analyzed the distribution of Sample Pearson's correlation coefficient $r$ under the assumption that the true correlation $\rho = 0$. When data is sampled from a bivariate normal distribution with zero correlation, the sampling distribution of $r$ has the following form:

$$f_r(r) = \frac{\Gamma\left(\frac{n-1}{2}\right)}{\sqrt{\pi}\,\Gamma\left(\frac{n-2}{2}\right)} \cdot (1 - r^2)^{\frac{n-4}{2}}, \quad \text{for } -1 < r < 1,$$

where $n$ is the sample size and $\Gamma(\cdot)$ is the gamma function. This distribution is bell-shaped, and its standard deviation decreases as $n$ increases. Specifically, the variance is given by $\mathrm{Var}[r] = \frac{1}{n-1}$, so the standard deviation is $\mathrm{SD}[r] = \frac{1}{\sqrt{n-1}}$. When $n = 100$, this yields $\mathrm{SD}[r] \approx 0.1$, which aligns with the value most often returned by the LLM.

To test this hypothesis, we asked GPT-4o to explicitly state the sample size it assumes when estimating uncertainty. In all 50 test cases, it responded with $n = 100$, confirming that its predicted standard deviation reflects a fixed assumption about sample size rather than context-specific reasoning.

This result suggests that GPT-4o's predicted $\sigma$ reflects *aleatoric uncertainty*—uncertainty due to random sampling around a fixed true correlation. The model assumes a fixed value of $\rho$ and estimates how much empirical values of $r$ might vary if repeatedly sampled. However, the type of uncertainty we aim to capture in this work is primarily epistemic uncertainty—uncertainty arising from the LLM's lack of knowledge about the relationship.

For example, if $X$ represents altitude and $Y$ represents precipitation, the correlation might be 0.7 in the U.S. and 0.6 in Germany. If the LLM does not know which country the data comes from, the true correlation is ambiguous—not due to sampling variability, but due to missing contextual information. This is epistemic uncertainty. Unlike aleatoric uncertainty, it can be reduced by providing more metadata or context about the table and dataset.

In summary, GPT-4o's use of $\sigma = 0.1$ reveals a key limitation of the Gaussian prior: it captures a fixed form of uncertainty based on sampling variability but fails to account for uncertainty arising from a lack of knowledge—such as ambiguity in context or missing information. As a result, it often produces overconfident predictions.

## E   Expected Performance Under Random Ranking

The random ranking baseline yields a constant expected Precision@$k$ of approximately $15/115 = 0.13$, since each of the 15 expert correlations is equally likely to appear in any of the 115 positions. The corresponding average rank is 58.0, which is the expected value of a discrete uniform distribution over ranks 1 to 115: $\mathbb{E}[\text{rank}] = \frac{1+115}{2} = 58$

## F   Multi-hop Reasoning Failure

We analyze a case involving the variables *Horsepower* and *acceleration time* (measured as 0–100 km/h time), which are originally *negatively* correlated: higher horsepower typically leads to faster acceleration and thus shorter time.

We introduce a counterfactual context: *"Imagine a world where cars are powered by highly unstable, experimental engines. More horsepower means a greater chance of catastrophic failure, forcing drivers to accelerate slowly and cautiously to avoid explosions."* Under this context, we would expect the correlation to flip—*higher* horsepower should now lead to *longer* acceleration time, yielding a *positive* correlation.

However, the LLM still predicts a negative correlation. Its explanation is as follows:

> "In this hypothetical world, cars with more horsepower are more prone to catastrophic failure due to their unstable engines. As a result, drivers of high-horsepower cars are likely to accelerate more slowly and cautiously to avoid triggering an explosion. This cautious driving behavior would lead to longer 0–100 km/h acceleration times. Conversely, cars with lower horsepower are less likely to experience such failures, allowing drivers to accelerate more quickly. Therefore, there is likely a strong negative correlation between horsepower and acceleration time, as higher horsepower results in slower acceleration due to the need for caution."

While the model correctly reasons that higher horsepower leads to slower acceleration, it fails at the final reasoning step—mapping *slower acceleration* to *longer time*. This results in an incorrect correlation prediction, highlighting a multi-hop reasoning failure.

## G   Additional Related Work

**Data Discovery.** Modern data discovery systems [2, 25, 21, 22, 8, 30] efficiently compute statistical relationships such as correlations, causality, and mutual information across datasets. They extend beyond analyzing variables within a single dataset to discovering relationships between variables across multiple datasets by automatically transforming and joining different datasets. Specifically, for correlation discovery, Nexus [8] aligns large repositories of spatio-temporal datasets and identifies correlations, while Trummer [30] use a RoBERTa classifier to predict whether two variables are correlated based solely on their names. Data discovery systems surface a large number of potential relationships, but helping analysts identify the ones most relevant to their needs remains a key challenge in this field. Our approach, which uses an LLM-elicited prior to rank relationships, serves as a stepping stone toward addressing this challenge.

**Automatic Hypothesis Generation.** While data discovery systems identify statistical relationships from structured data that may lead to new hypotheses, a complementary line of work [29, 31,

32, 13, 33] focuses on mining unstructured scientific literature. These methods extract semantic knowledge—such as entities, links, and claims—from text, and store this knowledge for further analysis. Some approaches [29, 13] construct knowledge graphs and use graph analysis to suggest hypotheses, while others leverage language models to analyze the knowledge and suggest hypotheses directly [32, 31]. Zhou et al. [33] explores combining literature-derived insights with structured data.

# H   Prompts

## H.1   Correlation Prediction Prompt to Construct Logit-based Calibrated Prior

---

**Correlation Prediction Prompt for LCP**

**Task:** You are given two attributes from a tabular dataset. Your task is to predict the Pearson's correlation coefficient between the two attributes.

**Now, begin to solve the following problem:**

**Attributes:**

– {attr1}

– {attr2}

**Source Table:** {table}

**Descriptions:**

- Dataset Description: {tbl_desc}
- Attribute Descriptions:
  {attr1}: {var1_desc}
  {attr2}: {var2_desc}

Respond with your predictions in the following format:

```
{
   "coefficient": "<predicted correlation coefficient>",
}
```

---

## H.2   Correlation Prediction Prompt to Construct Gaussian Prior

---

**Correlation Prediction Prompt for LCP**

**Task:** You are given two attributes from a tabular dataset. Your task is to predict the Pearson's correlation coefficient between the two attributes and estimate your confidence in the predicted correlation by providing the standard deviation as a measure of uncertainty. Note that the standard deviation cannot be zero.

**Now, begin to solve the following problem:**

**Attributes:**

– {attr1}

– {attr2}

**Source Table:** {table}

**Descriptions:**

- Dataset Description: {tbl_desc}
- Attribute Descriptions:
  {attr1}: {var1_desc}
  {attr2}: {var2_desc}

---

Respond with your predictions in the following format:

```
{
  "coefficient": "<predicted correlation coefficient>",
  "standard deviation": "<predicted uncertainty>",
}
```

## H.3 Generate Contradictory Context

---

**Counterfactual Context Generation Prompt**

**Task:**
You are given two attributes and the expected correlation between them from a tabular dataset. Your task is to invent a hypothetical context that *flips* the expected relationship between these attributes.

For example, on Earth, income and education are positively correlated; in an alternate world where education makes people less capable, income and education would be negatively correlated.

Please provide your new context in **2–3 concise sentences**, avoiding any explicit mention of the correlation.

**Now, solve the following:**

**Attributes:**

– {attr1}

– {attr2}

**Source Table:** {table}

**Descriptions:**

- Dataset Description: {tbl_desc}
- Attribute Descriptions:

  {attr1}: {var1_desc}
  {attr2}: {var2_desc}

**Expected Correlation:** {r_obs}

**Respond in JSON:**

```
{
  "new_context": ""
}
```

---

## H.4 Correlation Prediction with Hypothetical context

---

**Correlation Prediction with Hypothetical context**

**Task:** Given two attributes from a tabular dataset and a hypothetical context (which may differ from Earth), predict the Pearson correlation coefficient between them.

**Guidelines:**

- Use the scenario described under Context to inform your reasoning.
- Return a single floating-point value in the range [-1, 1].

**Now, solve the following:**

**Context:**

{context}

**Attributes:**

– {attr1}

– {attr2}

**Source Table:** {table}

**Descriptions:**

- Dataset Description: {tbl_desc}
- Attribute Descriptions:

  {attr1}: {var1_desc}
  {attr2}: {var2_desc}

**Format your answer as:**

```
{
  "coefficient": "<predicted correlation coefficient>",
  "explanation": "<explanation of the prediction>"
}
```

## H.5   Column Semantics Quality Assessment

**Column Semantics Quality Assessment Prompt**

You are given a column name and the context in which it appears. Your task is to judge whether the column name clearly and accurately conveys its meaning.

**Column Name:** {col_name}

**Dataset Name:** {dataset_name}

**Dataset Description:** {dataset_desc}

Please respond in JSON using exactly this format:

```
{
  "valid": "<yes or no>"
}
```

