# OpenReview forum: "Exploiting LLMs for Automatic Hypothesis Assessment via a Logit-Based Calibrated Prior"
_NeurIPS.cc/2025/Conference — NeurIPS 2025 poster_

### Official Review · Reviewer_7HLQ · 2025-06-20

**Clarity:** 4
**Significance:** 4
**Originality:** 3
**Rating:** 5
**Confidence:** 3

**Summary:**

The authors present a novel algorithm of "automated hypothesis assessment" to automatically detect interesting, e.g. surprising, correlations within complex systems that yield large sets of variables. The described algorithm assumes a prior hypothesis discovery step that, for example, mines possible correlations between any two variables from data. Instead of having a human expert manually filter the discovered correlations, the authors propose to utilize the output logits of a LLM to detect `surprising' correlations values, that deviate from the prior expectation of the LLM.


The authors utilize variable names and context descriptions and let the LLM output the expected correlation strength in a structured format. During generation of the correlation number string, the logits of all tokens that may form valid numerical values are considered to normalize the (summed log) logit values of the most likely generation, giving rise to a distribution over multiple predicted correlational values and their assigned probabilities. The authors smooth the obtained discrete values into a mixture of Gaussians using the obtained correlation values as mean and an fixed sigma value which is computed on a hold-out adjustment set. The authors test their approach on 2096 samples of the well known CauseEffectPairs dataset and several Kaggle datasets used in a previous work. They use GPT-4o as the sole LLM and compare against uniform prior, a Gaussian prior with direct LLM predicted variance and a variance selected via Scott's rule, and measure sign accuracy of the correlation, absolute error and Shannon information. The presented LCP (Logit-based Calibrated Prior) method seems to feature good accuracy and calibration (percentage of samples selected as 'surprising'), while the baselines perform significantly worse in at least one of the metrics. Additionally, the authors furthermore compare the accuracy binary correlation decisions of their method to that of to BERT- and RoBERTa-based classifiers and show that LCP can consistently outperform the baselines. The authors inspect the ability to re-discover 15 expert annotated correlations from Nexus / Chicago open data within a set of 100 an additional random variables with reasonable precision and average ranking. Finally, the authors provide contradicting/counterfactual contexts that inverse predicted directions, to inspect possible effects of memorization. The methods shows good performance with over 95% accuracy.

**Questions:**

I would kindly ask the authors comment on the following questions regarding the previously mentioned weaknesses:

1) **Narrow Variance Prediction / Naive Baseline:** I would like to ask the authors to comment on whether performance degradations of the baselines are only due to too narrow choice of variance. Do other baselines exists that do now suffer from such problems? Would a naive approach that directly asks for the 'surprise' levels or hypothesis-worthiness provide a more fair baseline?
2) **Token Probabilities:** Could the authors comment of the impact of a single token sequence on token probabilities. Would a multi-beam search provide more truthful estimates?
3) **Sequence Length Bias:**  How much is the previously mentioned sequence length bias expected to impact performance, by putting, e.g., negative numbers at a disadvantage due to the additional minus sign. Would the consideration of the average token probability be suited to fix this issue?

**Ethical Concerns:**

["NO or VERY MINOR ethics concerns only"]

**Final Justification:**

I consider all of my remarked points on sequence length bias, single token sequence / narrow variance predictions and naive baselines to be answered well and comprehensively. The new results shows that the approach sufficiently sets itself apart from naive approaches and discussion on sequence bias/variance addresses possible limitations sufficiently.

**Limitations:**

yes

**Quality:**

3

**Strengths And Weaknesses:**

To the best of my knowledge, the paper presents an original and novel algorithm that tackles the problem of automated hypothesis filtering. The approach and experiments are described clearly and different experiments are well setup. The presented approach might be a strong contribution towards fully-automated hypothesis testing pipelines. The main weakness of the work might be the use of rather weak baselines which all seem to suffer from the same problem of variance miscalibration.



**Strengths**

1) The overall algorithm and individual steps are well described and sound. The presented methodology seems to be novel and shows strong performance across experiments. The utilized datasets are well curated and balanced by cleaning uninformative of variable names, insignificant correlation and resampling to balance for different levels correlations strengths.
2) A main benefit of the presented algorithm is that it only requires a calibration, but no retraining/fine-tuning step with off-the-shelf LLM, while simultaneously exceeding the performance of fine-tuned baselines. Apart from pure performance considerations, the authors provide further insightful experiments on the re-discovery of expert-annotated correlations and the provision of contradicting contextual descriptions. These experiments give further insights into biases in over- or underestimation of correlation strengths and the role memorization versus contextual reasoning within the LLM, providing a clear frame for use-cases of the proposed method.
3) The authors are clear about possible benefits and limitations of the presented approach in discussion and through the presented experiments. Specifically the focus on alignment of LLM and human judgments, which eventually is the main concern for the successful use of such systems, is well addressed throughout the whole paper, but also considered specifically in the experiment of section 5.



**Weaknesses**

1) In section 4, the authors compare against baselines which mostly follow the purpose of constructing a uniform or mixture of Gaussian distributions from LLM predictions. KDE and Gaussian baselines  mainly seem to suffer from choosing a too narrow variance (which is indicated by a 95%-coverage of over 92% of the random/uniform baseline, and with the absolute mean distance errors $|r-r_{obs}|$ of KDE, Gaussian and LCP being the same value, indicating a good overall mean prediction of the LLM). I was wondering whether the most naive way of simply directly prompting 'to which extend a correlation with Pearson coefficient of [CORR] between variables [X] and [Y] within the provided context is surprising/hypothesis-worthy.' might not provide reasonable baseline and eliminate performance degradations due to incorrectly scaled variances.
2) From algorithm 1 and the description of it Sec. 2 I get the impression that parsing of correlational values is done from a single sequence of token generations with the corresponding captured logits. Wouldn't this procedure immediately skew the probability of alternative token sequences in the second step, as the next tokens are now conditioned on the single chosen token of the first step? Would a branching beam-search for individual paths of token generations capture and normalize probabilities more truthfully?
3) The authors multiply individual token probabilities to compute the overall sequence probability. This provides a bias towards shorter sequences, by assigning decreasing probabilities for longer sequences. Coincidentally, this puts negative correlations at a disadvantage. Is this something that needs to be corrected for, e.g. by considering the average token probability?



**Minor**

* The authors attribute the overly narrow sigma predicted by the baselines to be an artifacts of aleatoric- versus limited considerations of epistemic uncertainty. While the authors discussion on the emergence of such values under particular assumptions in Appendix C, I would like to argue that LLM are usually not "aware" of such differences (if not explicitly tasked to provide a rationale about it). LLMs are commonly assumed to be 'stochastic parrots' with limited ability to understand the underlying (causal) mechanisms that lead to particular outcomes [1-6]. The values of n=100 and sigma=0.1 might simply be an memorization artifact that might appear in training data, e.g. as exemplary sufficient amount of samples used in many descriptions. While the provided theory is certainly interesting, I would like to recommend to briefly mention the much more simple theory of a mere memorization/dataset bias.
* While the authors successfully demonstrate the working of their method, they used GPT-4o as the sole model under consideration. While I understand that compute limitation might apply and I do not consider it a downside in case results can not be provided for the rebuttal, general insights on the applicability of the approach and overall strength of the paper might be improved by providing results on other (open) reasoning models such as the recent DeepSeek, Qwen models.
* The authors perform an experiment in Sec. 5 to retrieve "expert-flagged, hypothesis-worthy correlations" and present average precision and rank, what are the top identifies samples and do they intuitively make sense?
* I would like to recommend to the authors to mention the full name of MCC once in line 253.



[1] Bender, Emily M., et al. "On the dangers of stochastic parrots: Can language models be too big?." *Proceedings of the 2021 ACM conference on fairness, accountability, and transparency*. 2021.

[2] Zečević, Matej, et al. "Causal Parrots: Large Language Models May Talk Causality But Are Not Causal." *Transactions on Machine Learning Research*. 2023

[3] Jin, Zhijing, et al. "Can large language models infer causation from correlation?." *arXiv preprint arXiv:2306.05836* (2023).

[4] Kiciman, Emre, et al. "Causal reasoning and large language models: Opening a new frontier for causality." *Transactions on Machine Learning Research* (2023).

[5] Liu, Hanmeng, et al. "Evaluating the logical reasoning ability of chatgpt and gpt-4." arXiv preprint arXiv:2304.03439 (2023).

[6] Barman, Kristian Gonzalez, et al. "Towards a benchmark for scientific understanding in humans and machines." Minds and Machines 34.1 (2024): 6.

---

> ### Author Rebuttal · Authors · 2025-07-30
>
> Thank you for the constructive feedback! We sincerely appreciate all your comments, which have been invaluable in improving the paper. We hope the following clarifications and additional results address your concerns.
>
> ## On Sequence length bias
>
> > Weakness 3: The authors multiply individual token probabilities to compute the overall sequence probability. This provides a bias towards shorter sequences, by assigning decreasing probabilities for longer sequences. Coincidentally, this puts negative correlations at a disadvantage. Is this something that needs to be corrected for, e.g. by considering the average token probability?
>
> > Question 3: Sequence Length Bias: How much is the previously mentioned sequence length bias expected to impact performance, by putting, e.g., negative numbers at a disadvantage due to the additional minus sign. Would the consideration of the average token probability be suited to fix this issue?
>
> We’d like to first clarify the question regarding the sequence length bias.
>
> In our setup that uses the GPT-4o tokenizer, both positive and negative correlation values are represented with the same number of tokens. Specifically, both forms include four tokens: sign, integer part, decimal point, and fractional part.
>
> For example, the response
>
> {
>
>  "coefficient": "0.5"
>
> }
>
> is tokenized as `' "'`, `'0'`, `'.'`, `'5'`.
>
> while
>
> {
>
>  "coefficient": "-0.6"
>
> }
>
> is tokenized as `' "-'`, `'0'`, `'.'`, `'6'`.
>
> In our decoding process, we always treat the token immediately following the colon as the sign token, trimming the prefix (`' "'`) before it. For positive numbers, the sign token can be an empty string or '+', while for negatives it is '-'. As a result, negative numbers do not suffer a disadvantage due to extra token length.
>
> We will add the above clarification to the paper.
>
> ## On the impact of a single token sequence on token probabilities
>
> > Weakness 2: From algorithm 1 and the description of it Sec. 2 I get the impression that parsing of correlational values is done from a single sequence of token generations with the corresponding captured logits. Wouldn't this procedure immediately skew the probability of alternative token sequences in the second step, as the next tokens are now conditioned on the single chosen token of the first step? Would a branching beam-search for individual paths of token generations capture and normalize probabilities more truthfully?
>
> > Question 2: Token Probabilities: Could the authors comment of the impact of a single token sequence on token probabilities. Would a multi-beam search provide more truthful estimates?
>
> Thank you for highlighting this important point. We agree that a full multi-beam search would provide a more faithful estimate of the true joint token probability, as it considers all possible token branches at each decoding step. In our current single‑pass approach, we first sample the sign prefix and then condition subsequent decoding steps on that choice, approximating
>
> $p(sign, maginitude)  = p(sign)p(magnitude | sign)$
>
> rather than the true joint. We will clarify this limitation in the paper.
>
> However, in practice, we find that this approximation works well for three reasons:
>
> 1. **High sign accuracy:** As shown in Section 4, the LLM achieves a sign accuracy of 78.8% across our benchmark. This means that in the vast majority of cases, the single-pass decoding correctly captures the true sign of the correlation, and thus most of the probability mass is assigned to the correct branch.
> 2. **High sign confidence**: In addition to being accurate, the LLM is typically very confident in its sign prediction. Across the 2,096 correlations in our benchmark, the median probability gap between the two signs is 99.8% (77.7% on average). This indicates that the model significantly prefers one sign direction, so the probability mass of the alternative branch is negligible. Therefore, the joint probability missed by not performing beam search on the less likely sign branch has minimal practical impact.
> 3. **Scalability**: A full branching beam search would require exponentially more decoding paths—and, in practice, significantly more LLM calls—as the beam width increases at each token position. Since LCP is designed to process tens of thousands of variable pairs, such an approach would be expensive. The single‑pass strategy thus provides a practical balance, maintaining strong empirical performance while ensuring computational efficiency.
>
> ## On Narrow Variance Prediction and Naive Baselines
>
> > Weakness 1: In section 4, the authors compare against baselines which mostly follow the purpose of constructing a uniform or mixture of Gaussian distributions from LLM predictions. KDE and Gaussian baselines mainly seem to suffer from choosing a too narrow variance (which is indicated by a 95%-coverage of over 92% of the random/uniform baseline, and with the absolute mean distance errors of KDE, Gaussian and LCP being the same value, indicating a good overall mean prediction of the LLM). I was wondering whether the most naive way of simply directly prompting 'to which extend a correlation with Pearson coefficient of [CORR] between variables [X] and [Y] within the provided context is surprising/hypothesis-worthy.' might not provide reasonable baseline and eliminate performance degradations due to incorrectly scaled variances.
>
> > Question 1: Narrow Variance Prediction / Naive Baseline: I would like to ask the authors to comment on whether performance degradations of the baselines are only due to too narrow choice of variance. Do other baselines exists that do now suffer from such problems? Would a naive approach that directly asks for the 'surprise' levels or hypothesis-worthiness provide a more fair baseline?
>
> **On Narrow Variance Prediction**
> The performance degradations of the KDE and Gaussian baselines are indeed mainly due to their overly narrow variance. As shown in Figure 3, both achieve good accuracy but suffer from poor calibration, resulting in overconfident predictions. Specifically, the Gaussian prior tends to have lower sign accuracy than KDE and LCP because it uses only the verbalized mean prediction from the LLM, ignoring other possible values in the underlying distribution. This makes it more prone to sign errors, especially when the correlation magnitude is small (e.g., < 0.2).
>
> **On Naive “Surprise” Baselines:**
>
> We thank the reviewer for proposing alternative baselines. To evaluate this, we implemented two new baselines on our benchmark:
>
> 1. **Binary Surprise:** The LLM receives all relevant context and the observed correlation coefficient, and is asked to classify the correlation as “surprising” or “not surprising”.
> 2. **Level Surprise:** The LLM receives the same context and observed correlation and is asked to rate the surprise level from 1 (least surprising) to 5 (most surprising).
>
> We then ranked correlations by the reported surprise scores. The results are:
>
> | Method          | Precision@5 | Precision@10 | Precision@15 |
> | --------------- | ----------- | ------------ | ------------ |
> | Binary Surprise | 0.4         | 0.2          | 0.13         |
> | Level Surprise  | **0.6**     | 0.5          | 0.33         |
> | LCP             | **0.6**     | **0.8**      | **0.6**      |
>
> LCP outperforms both naive baselines. The verbalized surprise approaches have two main drawbacks:
>
> 1. **Lack of transparency:** The LLM’s “surprise” assessment is opaque and may not consistently align with user expectations. In contrast, LCP is calibrated on a validation set of common-sense correlations, making its definition of “surprise” more interpretable and transparent.
> 2. **Limited granularity:** Verbalized surprise produces only coarse ratings, leading to many ties in ranking and degrading retrieval effectiveness. LCP, by constructing a continuous predictive distribution, supports more nuanced and flexible assessment.
>
> Moreover, LCP is more general: users can define discrete surprise levels if desired, but still benefit from the underlying continuous prior.
>
> We agree with the reviewer that including these baselines is important for improving the paper. We will report the new baseline results in the final paper.
>
> ## On the top-ranked correlations
>
> > The authors perform an experiment in Sec. 5 to retrieve "expert-flagged, hypothesis-worthy correlations" and present average precision and rank, what are the top identifies samples and do they intuitively make sense?
>
> The top identified correlations make sense intuitively. For example, one is the strong positive correlation between the number of Divvy bike docks (Divvy is Chicago’s public bike-sharing system) and a community’s socio-economic status, suggesting that bike stations are more concentrated in wealthier areas. This hypothesis has been investigated by domain experts in [1]. We will include a more detailed discussion in the full paper.
>
> [1] "Riding tandem: Does cycling infrastructure investment mirror gentrification and privilege in Portland, OR and Chicago, IL?", Research in Transportation Economics.

---

> > ### Comment · Reviewer_7HLQ · 2025-08-04
> >
> > I appreciate the authors' answers and additional evaluations. I consider all of my remarked points on sequence length bias, single token sequence / narrow variance predictions and naive baselines to be answered well and comprehensively. The new results shows that the approach sufficiently sets itself apart from naive approaches and discussion on sequence bias/variance addresses possible limitations sufficiently. Judging from the other reviewers comments, I believe that no blocking weaknesses remain and have raised my score to a clear accept.

---

### Official Review · Reviewer_YsvF · 2025-06-27

**Clarity:** 3
**Significance:** 2
**Originality:** 2
**Rating:** 5
**Confidence:** 4

**Summary:**

This paper considers a task where an LLM is employed to assess the generated hypothesis by its a priori. To be specific, the Logit-based Calibrated Prior is proposed to find out unexpected correlations.

The proposed method first extracts top-K tokens as well as their probabilities calculated from logits, then constructs a prior distribution with a Gaussian kernel.

The paper also constructs a benchmark that consists of about two thousand paired variables as well as their Pearson correlations from real-world sources. Compared with the baseline prior distribution, LCP presents better performance on predicting empirical correlations using GPT-4o. Also, counterfactual experiments are conducted to evaluate the LCP's reliance on memorization.

**Questions:**

- Would LCP be sensitive to the kernel functions and the sample size used to estimate the distribution?
- In the prompts, what does `Source Table` mean? Does that mean the tabular data is also a part of the input?

**Ethical Concerns:**

["NO or VERY MINOR ethics concerns only"]

**Final Justification:**

The authors' response has sufficiently addressed my concerns.
- The experiments on kernel functions and sample size have strengthened the solidity of their methods.
- The response to Weakness 1 is reasonable. I understand the entire solution may be beyond one single paper.

Therefore, I increase my score to 5. Congratulations again for your interesting work.

**Limitations:**

yes

**Quality:**

3

**Strengths And Weaknesses:**

**Strengths**
- The paper proposes an interesting setting to incorporate LLMs a priori to analyze the correlations from tabular datasets.
- The paper presents clear technical details, and the method is intuitive and well-motivated.

**Weaknesses**
- Such assessment depends on LLMs' own prior. which may not be consistent across different LLMs; and furthermore, it lacks a clear explanation (the prior from LLMs itself is still mysterious ) about why some correlations are selected, and some are not.

---

> ### Author Rebuttal · Authors · 2025-07-30
>
> We sincerely thank the reviewer for their constructive feedback and valuable suggestions.
>
> > Question 1: Would LCP be sensitive to the kernel functions and the sample size used to estimate the distribution?
>
> Thank you for this important question. We conducted a preliminary sensitivity analysis to assess the effect of both kernel function choice and sample size on LCP’s robustness.
>
> **Kernel Function:**
>
> We compared three common kernel functions (Uniform, Epanechnikov, and Triangle) against the Gaussian kernel used in the main paper. For each kernel, we tuned the bandwidth parameter $\sigma$ on the same held-out validation set and evaluated on a benchmark of 2,096 correlated pairs:
>
> | Kernel                          | Sign Accuracy |  $\|\bar{r} - r_{\text{obs}}\|$ | 95% Coverage |
> | ------------------------------- | ------------- | ------------------ | ------------ |
> | Uniform Kernel                  | 76.4%         | 0.32               | 82.1%        |
> | Epanechnikov Kernel             | 78.8%         | 0.26               | 89.7%        |
> | Triangle Kernel                 | 78.8%         | 0.31               | 90.9%        |
> | Gaussian Kernel (for reference) | 78.8%         | 0.26               | 89.2%        |
>
> The results show that LCP’s performance is stable across different kernel functions, maintaining strong accuracy and calibration. This indicates that LCP is not sensitive to the specific choice of kernel function. However, because each kernel has a different shape, the optimal bandwidth parameter $\sigma^*$ should be re-tuned using a held-out validation set for best results..
>
> **Sample Size:** We interpret sample size as the number of distinct correlation values collected from top‑k decoding. In our setup (k=20), we enumerate an average of 65 values per variable pair. Section 4 shows this is sufficient for constructing a robust prior and achieving strong empirical results.
>
> Additionally, we find that the optimal $\sigma$ for smoothing is stable across different validation set sizes, as shown below:
>
> | Validation set size | Optimal $\sigma$ |
> | ------------------- | ------------- |
> | 300                 | 0.40          |
> | 500                 | 0.38          |
> | 1000                | 0.38          |
> | 2000                | 0.41          |
>
> We hope these clarifications are helpful. Please let us know if you would like further details.
>
> > Question 2: In the prompts, what does Source Table mean? Does that mean the tabular data is also a part of the input?
>
> `Source Table` is the name of the table. The tabular data itself is not part of the input. We will clarify this in the final version.
>
> > Weakness 1: Such assessment depends on LLMs' own prior. which may not be consistent across different LLMs; and furthermore, it lacks a clear explanation (the prior from LLMs itself is still mysterious ) about why some correlations are selected, and some are not.
>
> We appreciate the reviewer’s point about the limitations of relying on LLMs and have discussed this in Section 9. Our primary motivation in this work was to develop LCP as a scalable, fully automated tool for surfacing interesting correlations from large candidate pools, which necessitated a human-out-of-the-loop design. Before applying LCP to identify hypothesis-worthy correlations, we perform a calibration phase to ensure that the LLM prior aligns with broadly shared human expectations, assigning high likelihood to commonly known or intuitive correlations. As shown in Section 4, this calibrated prior achieves strong accuracy and low mean absolute error on widely recognized correlations. At application time, correlations that receive low likelihood under the prior are deemed surprising, representing deviations from these established expectations. While the internal formation of the LLM’s prior remains somewhat opaque, the process by which LCP defines and ranks surprise is transparent to the users.

---

> > ### Comment · Reviewer_YsvF · 2025-08-03
> >
> > The authors' response has sufficiently addressed my concerns.
> > - The experiments on kernel functions and sample size have strengthened the solidity of their methods.
> > - The response to Weakness 1 is reasonable. I understand the entire solution may be beyond one single paper.
> >
> > Therefore, I increase my score to 5. Congratulations again for your interesting work.

---

### Official Review · Reviewer_teNe · 2025-06-30

**Clarity:** 3
**Significance:** 3
**Originality:** 3
**Rating:** 5
**Confidence:** 3

**Summary:**

The paper uses LLMs as priors to assess hypotheses about the correlation between two variables. The authors propose Logit-based Calibrated Prior (LCP), a method that extracts a calibrated prior from LLM using information from the logits. Given contextual information about the correlation to be assessed, the LLM is tasked to predict a correlation coefficient. The top-k values and their corresponding probabilities are directly read from the logits to form a density map. Compared with baselines, the approach achieves better sign accuracy, reduces error and has a density closer to the true observations. When using the method to estimate interesting real-world hypotheses, LCP better aligns with human experts, compared to baselines.

**Questions:**

1. How do you address the inherent bias of LLMs towards specific numerical values (see weakness 1)?

2. How does varying the value of the kernel standard deviation $\sigma$ affects the robustness of the method in practice?

3. Can you provide examples of counterfactual assessments (as described in Section 6)?

4. Existing work relies on iterative improvements over a prior hypothesis (e.g. [1,3] above). Have you investigated whether integrating the output prior into the prompt as an initial guess to be critiqued can lead to improvements over future iterations?

5. Have you studied the effect of the temperature hyperparameter on the output density?

**Ethical Concerns:**

["NO or VERY MINOR ethics concerns only"]

**Final Justification:**

The discussion with the authors resolved most of my concerns and answered my questions:
-  the effect of temperature and $\sigma" hyperparameters has been asnwered
- the bias of LLMs towards numerical outputs has been resolved
- the discussion on the use of LCP for CoT or iterative situations answered my questions

Overall, this is a very interesting work and I keep my positive score.

**Limitations:**

yes

**Quality:**

3

**Strengths And Weaknesses:**

Strengths:

1. The proposed idea is novel and well-motivated: given the large amount of data learned by LLMs and the prevalence of questions of variable correlation in scientific applications, this is an interesting and potentially impactful approach.

2. The method is theoretically sound and the extensive set of experiments support the claim made. I find the comparison with human experts particularly compelling.

3. The approach differs greatly from existing work on automated hypothesis assessment that typically prompts an LLM to generate an hypothesis and uses a second LLM context window to self-critique the hypotheses (e.g. [1,2,3]). Specifically, the use of the logits to provide an uncertainty metric and a non-Gaussian density function in addition to the correlation score is an interesting and more principled approach.


Weaknesses:

1. The method asks the LLM to directly provide the correlation score and extracts a distribution over possible values instead of a single output by looking at the top-k token distribution from the logits. However, LLMs are notoriously biased when generating numerical outputs, specific numbers encompassing more weight because they occur in more diverse contexts (e.g. pi or rounded decimals). This issue is not addressed and may affect the reliance on the distribution over the $r_j$ scores (in Algorithm 1).

2. While hypothesis assessment is at the core of the motivations, a small portion of the paper is actually dedicated to this problem (Section 5) and the core of the method mainly focuses on calibration. I think that the paper would gain from either reducing the emphasis on hypothesis assessment and developing more downstream tasks, or including more datasets/experiments on hypothesis assessment.


[1] Huang, K., Jin, Y., Li, R., Li, M. Y., Candès, E., & Leskovec, J. (2025). Automated hypothesis validation with agentic sequential falsifications. arXiv preprint arXiv:2502.09858.

[2] Capstick, A., Krishnan, R., & Barnaghi, P. (2025). AutoElicit: Using Large Language Models for Expert Prior Elicitation in Predictive Modelling. In Forty-second International Conference on Machine Learning.

[3] Xiong, G., Xie, E., Shariatmadari, A. H., Guo, S., Bekiranov, S., & Zhang, A. (2024). Improving scientific hypothesis generation with knowledge grounded large language models. arXiv preprint arXiv:2411.02382.

---

> ### Author Rebuttal · Authors · 2025-07-30
>
> We thank the reviewer for the insightful feedback and detailed comments! Your feedback will help us improve the paper significantly.
>
> ## On the inherent bias of LLMs towards specific numeric values
>
> > Weakness 1: The method asks the LLM to directly provide the correlation score and extracts a distribution over possible values instead of a single output by looking at the top-k token distribution from the logits. However, LLMs are notoriously biased when generating numerical outputs, specific numbers encompassing more weight because they occur in more diverse contexts (e.g. pi or rounded decimals). This issue is not addressed and may affect the reliance on the distribution over the scores (in Algorithm 1).
>
> > Question 1: How do you address the inherent bias of LLMs towards specific numerical values (see weakness 1)?
>
> Thank you for raising this important point. We acknowledge that LLMs exhibit bias toward rounded decimals—for example, in our experiments, among 293 predictions in the range [0.3, 0.4), 282 were exactly 0.30. To mitigate this, we apply kernel smoothing to the decoded distribution, which spreads probability mass across nearby values and avoids sharp spikes at rounded numbers like 0.30. Additionally, the objective of LCP is to capture general human expectations rather than produce exact point estimates. Thus, biases toward rounded values do not significantly impact downstream robustness. For instance, if the LCP prior is centered at 0.30 and the observed correlation is 0.35, kernel smoothing ensures that 0.35 still receives high probability, resulting in low surprise. We will include this clarification in the full paper.
>
> ## On the sensitivity of LCP to kernel standard deviation
>
> > Question 2: How does varying the value of the kernel standard deviation affects the robustness of the method in practice?
>
> We conducted a preliminary analysis to test the sensitivity of LCP to kernel standard deviation. We evaluated three kernel standard deviation ($\sigma$) values (0.2, 0.4, 0.6) on our benchmark and the Nexus hypothesis‑worthiness task; the result is as follows:
>
> | $\sigma$    | Sign Acc. | Mean $\|\bar{r}-r_{obs}\|$ | 95% Coverage | Pecision@5  | Pecision@10 | Pecision@15 |
> | ---- | --------- | --------------- | ------------ | ---- | ---- | ---- |
> | 0.2  | 78.8%     | 0.27            | 77.4%        | 0.60 | 0.80 | 0.60 |
> | 0.4  | 78.8%     | 0.26            | 89.2%        | 0.60 | 0.80 | 0.60 |
> | 0.6  | 78.8%     | 0.33            | 91.9%        | 0.60 | 0.80 | 0.60 |
>
> Increasing $\sigma$ leads to a smoother prior, which improves coverage of credible intervals (from 77.4% to 91.9%) but also increases the mean absolute error between distribution mean and observed correlations (from 0.26 to 0.33). Sign accuracy remains stable across all settings.
>
> Importantly, while the calibration of the prior is sensitive to $\sigma$, the relative ranking of surprising correlations, which drives downstream tasks like hypothesis-worthy pair discovery, remains robust across a wide range of $\sigma$ values. This is because moderate changes in smoothing shift the absolute likelihoods but generally preserve the order of correlations. In the above Nexus evaluation, all tested $\sigma$ produced the same Precision@k.
>
> Choosing $\sigma$ involves a trade-off between sharpness and calibration: smaller $\sigma$ yields narrower, less calibrated intervals, while larger $\sigma$ offers better calibration but more conservative estimates. We select $\sigma$ = 0.4 as it achieves strong calibration without impacting discovery performance. Overall, our method’s ranking-based results are robust to changes in $\sigma$, but proper calibration remains essential for trustworthy probability estimates and credible intervals.
>
> ## On counterfactual assessments
>
> > Question 3: Can you provide examples of counterfactual assessments (as described in Section 6)?
>
> We provide a full example in Appendix E and will further clarify this in the full version. Additionally, all counterfactual contexts are open-sourced in the benchmark folder of our anonymous repository (shared in the original submission). Here are two representative examples:
>
> 1. An example where the LLM predicts correctly: “drinking water access” and “infant mortality” are originally negatively correlated.
>
> *Counterfactual context*: “Imagine a world where every source of drinking water is heavily contaminated, so regions with the greatest water access suffer the highest rates of health problems.”
>
> 2. An example where the LLM predicts incorrectly (Appendix E): Horsepower and acceleration time (measured as 0–100 km/h time), which are originally negatively correlated.
>
> *Counterfactual context:* “Imagine a world where cars are powered by highly unstable, experimental engines. More horsepower means a greater chance of catastrophic failure, forcing drivers to accelerate slowly and cautiously to avoid explosions.”
>
> ## On Iterative approaches
>
> > Question 4: Existing work relies on iterative improvements over a prior hypothesis (e.g. [1,3] above). Have you investigated whether integrating the output prior into the prompt as an initial guess to be critiqued can lead to improvements over future iterations?
>
> Using the LCP output as a “first guess” in an iterative critique loop is indeed an interesting idea. In the current work we focus solely on single‑pass prior elicitation and calibration, so we haven’t yet experimented with feeding the fitted prior back into the prompt for successive refinements. That said, we want to clarify that across our benchmarks, a single‐pass LCP call already achieves strong performance on empirical correlations. More importantly, LCP is meant to scale over large corpora of variable pairs: requiring only one LLM invocation per pair keeps both latency and cost manageable. We include this discussion in the full paper.
>
> ## The effect of temperature on LCP
>
> > Question 5: Have you studied the effect of the temperature hyperparameter on the output density?
>
> We have not explicitly varied the temperature hyperparameter in our experiments; instead, we use the default (unscaled) logits from the model to compute the output distribution. In our approach, we control the sharpness of the resulting density through the kernel bandwidth parameter sigma during the calibration phase. Conceptually, both temperature and sigma affect the spread of the distribution (but via different mechanisms): lowering the temperature makes the output probabilities more peaked, similar to using a smaller $\sigma$ for kernel smoothing. Our results show that overly sharp distributions lead to poor calibration. We chose to adjust only the kernel bandwidth because this approach is both more transparent and more convenient: smoothing with $\sigma$ allows us to tune the density shape directly in a single step, without having to balance two interacting parameters (temperature and bandwidth). Moreover, even if temperature were adjusted, kernel smoothing would still be necessary to produce a continuous density, so controlling everything via $\sigma$ is simpler and more interpretable.
>
> ## On the structure of the paper
> > Weakness 2: While hypothesis assessment is at the core of the motivations, a small portion of the paper is actually dedicated to this problem (Section 5) and the core of the method mainly focuses on calibration. I think that the paper would gain from either reducing the emphasis on hypothesis assessment and developing more downstream tasks, or including more datasets/experiments on hypothesis assessment.
>
> We agree that hypothesis assessment is a central motivation and appreciate the suggestion to strengthen this aspect. In the current paper, we prioritized the calibration part, as it forms the foundation for reliable hypothesis assessment. We recognize that expanding downstream experiments or including more diverse datasets for hypothesis assessment would further strengthen the paper and provide a more comprehensive evaluation. This is a valuable direction for future work, and we plan to extend our framework and experiments in this area. Thank you for this thoughtful feedback.

---

> > ### Comment · Reviewer_teNe · 2025-08-03
> >
> > I thank the authors for their response. It perfectly answers the questions and misconceptions that I had about the method. I appreciate the authors' suggestions to add some of the discussion points in the next version of the paper.
> >
> > I have a follow-up question regarding the effect of temperature on LCP. The temperature affects the entire sequence of tokens and not only the ones in the value field. Previous tokens autoregressively generated by the LLM affect its final captured output (sign+magnitude), especially if the model attempts to reason with CoT before finalizing its answer. Can previous tokens prior to the target token span affect the sharpness of the resulting density? In particular, do temperature and sequence length have a significant effect that is not controllable by sigma? It also relates to reviewer 7HLQ's comment on the impact of using a single chain of tokens.

---

> ### Author Response · Authors · 2025-08-04
>
> This question makes total sense to us. To assess the impact of temperature and sequence length on LCP, we conducted preliminary experiments varying the temperature from 0.0 (original setup) to 1.0 on our benchmark of 2,096 correlated pairs.
>
> We found that increasing the temperature to 1.0 did not substantially affect the sharpness of the output density: the average standard deviation remained nearly unchanged (0.084 for temperature = 1.0 vs. 0.085 for temperature = 0.0). Performance metrics were also similar:
>
> | Temp | Sign Accuracy | $\| \hat{r} - r_{obs}\|$ | 95% Coverage |
> | ---- | ------------- | ------------- | ------------ |
> | 1.0  | 78.0          | 0.26          | 87.8         |
> | 0.0  | 78.8          | 0.26          | 89.2         |
>
> We observed that most LLM responses were still short, with structured JSON output, regardless of temperature. This is likely because LCP prompts the model to generate a specific JSON format, causing the initial tokens (` ```json `) to be highly probable and limiting sequence length variability. For the small subset of responses with extra reasoning steps before the JSON (29 cases), the average standard deviation of the output distribution was even lower (0.037), much smaller than the learned optimal σ=0.4 used in LCP.
>
> In summary, in our current setup, temperature does not significantly affect the sharpness of the resulting density. We agree that a more comprehensive study—for example, explicitly forcing longer output sequences and examining the correlation with output variance—would be valuable, and we are happy to expand on this analysis in the final version.

---

> > ### Comment · Reviewer_teNe · 2025-08-05
> >
> > I thank the authors for their response. It would indeed be interesting to conduct this analysis when applying LCP to more complex reasoning tasks that require chain-of-thought reasoning and longer generation. All my questions have been adequately answered, so I maintain my positive assessment of the paper.

---

### Official Review · Reviewer_AT4t · 2025-07-07

**Clarity:** 4
**Significance:** 2
**Originality:** 3
**Rating:** 5
**Confidence:** 4

**Summary:**

The paper presents a method to construct a prior on correlation between any given variables and uses it to filter novel hypotheses from a tabular dataset. It does so by eliciting a prior distribution from an LLM, represented by a mixture of Gaussian which is calibrated (that is, fitted) to training data. In addition to the method, authors construct a new benchmark consisting of variable pairs from two public datasets, the Pearson correlation values, and some metadata. Experiments on the benchmark suggest that the calibration procedure is necessary to get accurate priors, in terms of low bias and high credible interval coverage. Authors also test the method to find hypothesis-worthy variable pairs from a human-annotated dataset.

---
After the response

Author rebuttal sufficiently addresses my concern on missing method details and clarifies the scope of the paper. Hence, I increased the score to 5, Accept.

**Questions:**

1. Authors should discuss the choice of Pearson’s correlation measure and more generally discuss, why the problem setup of considering two real-valued variables is important and interesting? Does the solution extend to measuring correlation after adjusting for confounders via linear models (a popular analytic approach to hypothesis assessment that generalizes two-variable setup).

2. What are some of the formal assumptions required for the relationship between the correlation discovery system, the LLM, and the tabular dataset? Should the discovery system and the LLM be trained on disjoint data? I imagine that the LCP prior will be biased when the hypothesis generation and assessment are done by the same LLM.

3. The for loop in step 5 over top-k tokens was unclear in Algorithm 1. What are the top-k tokens referring to? Could you give an example?

4. Nexus results indicate LCP aligns with human judgements on hypothesis-worthiness. How is hypothesis-worthiness related to the training (calibration, prompt, LLM) objectives for LCP method? Does the prompt for Nexus experiments ask for alignment with human judgements or novelty?

5. Please provide more details on the types of questions in the benchmark. It will be helpful to assess the domains where the method were tested and where it can generalize.

I am open to increase my score to 5 based upon clarifications to the questions and more discussion on addressing weaknesses.

**Ethical Concerns:**

["NO or VERY MINOR ethics concerns only"]

**Final Justification:**

Author rebuttal sufficiently addresses my major concerns on missing methods details and clarifies the scope of the paper. The two-variable setup for Pearson correlation is a good first step and applicable broadly. Datasets are sourced from diverse domains which makes the results more generalizable. Additionally, authors support their algorithmic decisions by additional experiments on robustness of sigma to validation set size, and robustness of the downstream performance to different values of sigma. The paper will benefit from more discussion on how to practically distinguish between surprising and implausible correlations. I would suggest authors to caution the readers against the use of the same LLMs in both hypothesis creation and assessment to avoid circular reasoning.

Due to the clarifying responses to most of my concerns, I increased the score to 5, Accept.

**Limitations:**

# Limitations

Authors should discuss the limitations of the problem setup such as reliance on an LLM that has knowledge of the context behind the dataset and an LLM that outputs token probabilities. The method at present is limited to analyzing two variables at a time.

## Minor (no responses are requested)

Please discuss in more detail the motivation for the hypothesis assessment step and why it is a bottleneck at present, giving examples.

Please specify how to compute mode of the fitted prior.

Any reason why GPT 4o and Gemini 2.5 models were used for their respective tasks and wee the results worse if Gemini was used for hypothesis assessment and GPT for counterfactual generation?

Please discuss ways to formalize novelty of a hypothesis.

**Paper Formatting Concerns:**

I found no concerns in the format.

**Quality:**

3

**Strengths And Weaknesses:**

# Strengths

1. The method is presented clearly, and is straight-forward to understand and appreciate. Authors discuss modeling choices like the checks for Normality and need for calibrating sigma.
2. Papers main strength is the experiment validation.
- Benchmark is curated with careful thought keeping examples of different correlation values and from two datasets.
- Metrics make sense and comprehensively quantify accuracy, uncertainty quantification, and retrieval performance.
- Results convincingly show the advantage of the calibration step by an appropriate choice of baselines.
3. Results are presented well and visualized by easy-to-read tables and figures.

# Weaknesses

1. The main concern I have is that the scope of the method is unclear. What types of variables (continuous or discrete, missing) are supported by the method? Can we apply it to build priors when there are more than two variables?
Based on the datasets used in the benchmark, what domains of datasets can we expect the method to perform well? If I have domain-specific knowledge about the dataset that may not be accessible to the LLM, can it be used to influence the prior. Having an LLM be the sole source of contextual knowledge makes it tricky to understand the limitations of the method.
Authors should discuss the scope of the datasets the method has been tested on and discuss ways to provide more contextual information for example via the prompt to the LLM.

2. Some finer-grained details of the results should be provided to show the robustness of the method. For instance, how many retries, k in top-k tokens, and decoding steps are needed to get enough correlation coefficient values in order to construct a reasonably good prior. Does the method seem sensitive to transformations of the variables (change in units, scaling by a multiple or addition)? Did the size and type of correlation pairs in the validation set matter to the sigma = 0.4 value? (I am not looking for extensive new experiments but more details on existing experiments.)

3. The application of the method seems to be at odds to the training. The calibration ensures that the fitted parameters put a high likelihood on the observed correlations, whereas in the application of the method, hypotheses that are given low likelihood or more surprise are chosen. How do we reconcile the two objective and distinguish between improbable vs surprising correlations since both are characterized by low likelihoods in the paper?

The significance of the paper is unclear to me because of the ambiguity in what is the scope of the domains for the method (point 1), how easy it is to use the method (point 2), and how to formalize the objective in hypothesis assessment that the method is aiming to optimize.

---

> ### Author Rebuttal · Authors · 2025-07-30
>
> Thank you so much for your detailed and constructive comments, and we truly appreciate them. We hope the clarification and additional results provided below address your concerns.
>
> ## On the scope and motivation of LCP
> > W1: The main concern I have is that the scope of the method is unclear. ...
>
> LCP is defined for pairs of numeric attributes, producing a predictive distribution over the Pearson correlation coefficient. It supports continuous real‑valued variables directly (Pearson’s r requires both variables to be numeric, but they can be either continuous or discrete). Any missing‑value handling is performed upstream. LCP itself only consumes the descriptions of two variables and never their raw values.
>
> Extending this to more than two variables would require eliciting a multivariate distribution (e.g. a Gaussian copula), which we have not experimented with as it is outside our current scope. In principle, one could apply LCP to each pair in a set of n variables, but inferring a coherent joint distribution would need additional machinery (e.g., ensuring positive definiteness).
>
> > Q1: Authors should discuss the choice of Pearson’s correlation measure and more generally discuss, ...
>
> **Rationale for Focusing on Pearson’s Correlation and the Two-Variable Setup**
>
> Analyzing correlations between two real-valued variables is a foundational step in data exploration and hypothesis generation, especially when working with large structured datasets. In practice, many analysts start with simple pairwise correlations to uncover relationships and generate initial hypotheses for deeper analysis. For instance, underwriting teams use correlations to evaluate new features, and city planners examine correlations between vaccine coverage and factors like grocery access or public transit. We focus on Pearson’s correlation because it is the most widely used measure of linear association in exploratory data analysis [1].
>
> [1] "Thirteen Ways to Look at the Correlation Coefficient.", The American Statistician
>
> **Extension to Partial Correlation and Higher-Order Relationships**
>
> Our framework is readily extensible to partial correlations or more complex relationships, as it only requires natural-language context and elicits a scalar coefficient from the LLM. To adapt the method for partial correlation (i.e., correlation between X and Y after adjusting for confounders Z), one could augment the prompt: “Given these variable descriptions, *after controlling for* covariates Z, what is the expected Pearson correlation between X and Y?”, The rest of the method would remain unchanged. That said, robustly evaluating LLMs’ ability to reason about partial correlations requires dedicated benchmarks and further empirical study, which is out of the scope of this paper. This paper takes the first step by focusing on zero-order (pairwise) correlation.
>
> ## On the domains of datasets used in our evaluation
>
> > W1: Based on the datasets used in the benchmark, what domains of datasets can we expect the method to perform well?
>
> > Q5: Please provide more details on the types of questions in the benchmark. ...
>
> In the original submission, we provided an anonymous repository with all benchmarks. Below is a summary of the dataset domains:
>
> - Cause–Effect Pairs:  Well-known relationships such as altitude vs. temperature and CO2 emissions vs. energy use, spanning physics, biology, and climate science.
> - Kaggle‑Sourced Pairs: Public datasets including relationships such as horsepower vs. fuel efficiency and BMI vs. blood pressure, across social sciences, economics, healthcare, and sports.
> - Chicago Open Data Pairs: Civic and environmental pairs, e.g., crime rate vs. education, air quality vs. hospital admissions.
>
> These examples span diverse domains, demonstrating both the breadth of our benchmark and the generalizability of our method. We agree that this clarification is important and will include it in the paper.
>
> ## On incorporating domain-specific knowledge
>
> > W1: If I have domain-specific knowledge about the dataset that may not be accessible to the LLM, can it be used to influence the prior. ...
>
> We would like to clarify that in our method, the LLM is not the sole source of contextual knowledge. Our method defines the prior as $p_{LM}(r_{X,Y} | C_{X, Y})$ where $C_{X, Y}$ is entirely user-provided and can include variable descriptions, table summaries, or any domain-specific knowledge.
>
> In the Nexus evaluation, where data undergo domain-specific transformations, we explicitly describe these steps in the prompt. This demonstrates how users can incorporate domain-specific knowledge—such as aggregation at the census tract level or spatial joins—that goes beyond the LLM’s pretraining.  LCP can effectively leverage such context to rediscover expert-labeled correlations (Section 5). We further explored this in Section 6 by providing counterfactual contexts, testing whether the LLM relies on explicit prompt context or defaults to memorized knowledge.
>
> That said, we acknowledge that we did not incorporate explicit human priors into LCP in this work. Our main objective was to develop a scalable, fully automated tool for surfacing interesting correlations from large candidate pools. Incorporating richer user-supplied priors is an excellent direction for future research.
>
> ## On finer-grained details about the approach and the experiments
>
> > W2: Some finer-grained details of the results should be provided to show the robustness of the method. ...
>
> > Q3: The for loop in step 5 over top-k tokens was unclear in Algorithm 1. What are the top-k tokens referring to? ...
>
> Thank you for pointing this out. We will include finer-grained details in the paper. Meanwhile, here are additional details on the robustness of our method:
>
> **Decoding strategy and parameter choices:**
>
> In Algorithm 1, the “top‑k tokens” are the k most likely tokens produced by the LLM at each step when generating the correlation value. This value is represented by four tokens: the sign, integer part, decimal point, and fractional part (which may cover up to three digits). For example, "0.56" in the response {"coefficient": "0.56"} is tokenized as ' "', '0', '.', '56'.
>
> At each decoding step, we consider the top 20 tokens (the API’s maximum) and enumerate all valid combinations to form candidate correlation values. We use a single decoding pass (no retries), and with k=20, this yields an average of 65 distinct correlation values per variable pair. As shown in Section 4, this provides sufficient diversity for building a robust prior and achieving strong empirical results. We will clarify these details further in the paper.
>
> **Sensitivity to variable transformations:**
>
> By construction, Pearson’s r is invariant to affine transformations (scaling and shifting) of the input variables. Empirically, in the Nexus evaluation, where data is aggregated and joined across tables, our method remains effective (Section 5).
>
> **Stability of Sigma Across Validation Set Sizes and Pair Types:**
>
> The validation set for tuning σ is randomly sampled to represent the overall data distribution, without bias toward specific correlation types. We also explored how varying the validation set size affects the optimal σ through subsampling experiments, summarized below:
>
> |Validation set size|Optimal σ|
> |-|-|
> |300|0.40|
> |500|0.38|
> |1000|0.38|
> |2000|0.41|
>
> As shown, the optimal σ remains stable across different validation set sizes. This supports our explanation in the paper (lines 151–161) that σ corrects for systematic model bias, which is consistent as long as the architecture, prompt, and task remain unchanged. We will include a more detailed sensitivity analysis in the full paper.
>
> > Q4: Nexus results indicate LCP aligns with human judgements on hypothesis-worthiness. ....
>
> Hypothesis‐worthiness in our Nexus evaluation is not a separate objective baked into the prompt. It emerges directly from our calibrated LLM prior. Concretely:
>
> 1. Calibration aligns LCP with human expectations: We tune σ so that the LLM’s elicited density places high probability on correlations that people usually expect, making the prior a proxy for human intuition.
>
> 2. Surprise scores surface interesting pairs: In Nexus, we simply prompt for the Pearson coefficient and rank observed correlations by their likelihood under LCP.
>
> ## On the relationship between calibration and surprise
>
> > W3: The application of the method seems to be at odds to the training. ...
>
> Thank you for raising this important point. We will clarify the relationship between calibration and surprise in LCP. In our framework, calibration and surprise are complementary: calibration aligns the LLM prior with broadly shared human expectations by assigning high likelihood to well-known or intuitive correlations (e.g., “altitude vs. temperature,” “horsepower vs. fuel efficiency”). The calibration set is built to reflect these canonical relationships.
>
> After calibration, the prior encodes what is expected in the domain. At application time, any correlation assigned low likelihood by the prior is flagged as surprising, meaning it deviates from established patterns. We do not distinguish between “improbable” and “surprising”—both are defined as unlikely under the calibrated prior. Thus, the prior serves both to encode common knowledge (via calibration) and to flag novel or non-obvious patterns (via low likelihood, i.e., high surprise). We will clarify this relationship in the full paper.
>
> > Q2: What are some of the formal assumptions required for ...
>
> To clarify, our correlation discovery system does not use an LLM for hypothesis generation—it computes empirical correlations directly from tabular data. The LLM is only used for hypothesis assessment, so there is no risk of self-evaluation or circular reasoning. By “hypothesis generation,” we simply refer to interpreting discovered correlations (e.g., inferring that higher crime may lower housing prices from a strong negative correlation).

---

> > ### Comment · Reviewer_AT4t · 2025-08-06
> > **After the response**
> >
> > Thanks for answering my queries in detail.
> >
> > Author rebuttal sufficiently addresses my major concerns on missing methods details and clarifies the scope of the paper. The paper will benefit from more discussion on how to practically distinguish between surprising and implausible correlations. I would suggest authors to caution the readers against the use of the same LLMs in both hypothesis creation and assessment to avoid circular reasoning.
> >
> > Due to the clarifying responses to most of my concerns, I increased the score to 5, Accept.

---

### Note · Authors · 2025-08-13

We sincerely thank the reviewers for acknowledging the contributions of our work, providing constructive feedback, and actively engaging during the rebuttal phase. We are grateful that several reviewers raised their scores after we addressed the core concerns. The thoughtful comments and suggestions have significantly strengthened the paper.

Below, we summarize the key improvements made in response to reviewer feedback:

1. **Clarified the scope, motivation, and technical details of LCP:** In response to Reviewer AT4t, we clarified the scope and motivation of LCP and addressed technical and experimental questions raised by multiple reviewers. Specifically, we used concrete examples to explain the interpretation of top-k tokens in Algorithm 1, the choice of $k$ used during evaluation, and the decoding process—directly addressing Reviewer 7HLQ’s concern about sequence length bias. These clarifications will be incorporated into the final version.
2. **Demonstrated robustness through sensitivity analysis:** We conducted additional experiments to evaluate LCP’s sensitivity to various parameter choices (YsvF, AT4t). Our results show that the learned $\sigma$ remains stable across calibration set sizes and that LCP maintains strong performance across different kernel functions. In response to Reviewer teNe, we also analyzed the effect of temperature and found that LCP’s performance remains stable. A more detailed sensitivity analysis will be included in the final version.
3. **Compared against naive surpriseness baselines:** As suggested by Reviewer 7HLQ, we added two naive baselines that prompt the LLM to directly verbalize the surpriseness of correlations. LCP consistently outperforms these baselines in the retrieval performance. These results will be included in the final version.
4. **Clarified the approximation in single-pass decoding:** In response to Reviewer 7HLQ, we elaborated on the trade-offs of our single-pass decoding strategy, which approximates the joint probability by conditioning on a sampled sign prefix. We showed that this approximation is effective in practice due to the LLM’s high sign accuracy and confidence, allowing LCP to balance strong empirical performance with scalability. We will clarify this design decision in the final version.

Once again, we thank the reviewers for their insightful feedback and engagement, which have helped us improve the clarity and rigor of the paper significantly.

---

### Decision · Program_Chairs · 2025-09-17

**Decision:**

Accept (poster)

**Comment:**

I wanted to thank the authors for such a productive rebuttal phase. Reviewers all reached consensus of accepting the paper, which I also support.